# Computational identification of mutually exclusive transcriptional drivers dysregulating metastatic microRNAs in prostate cancer

Mengzhu Xue[1,*], Haiyue Liu[1,*], Liwen Zhang[2,3,4,*], Hongyuan Chang[1,3,*], Yuwei Liu[1], Shaowei Du[1,5], Yingqun Yang[1,3,4] & Peng Wang[1,2,3,4]

Androgen-ablation therapies, which are the standard treatment for metastatic prostate cancer, invariably lead to acquired resistance. Hence, a systematic identification of additional drivers may provide useful insights into the development of effective therapies. Numerous microRNAs that are critical for metastasis are dysregulated in metastatic prostate cancer, but the underlying molecular mechanism is poorly understood. We perform an integrative analysis of transcription factor (TF) and microRNA expression profiles and computationally identify three master TFs, AR, HOXC6 and NKX2-2, which induce the aberrant metastatic microRNA expression in a mutually exclusive fashion. Experimental validations confirm that the three TFs co-dysregulate a large number of metastasis-associated microRNAs. Moreover, their overexpression substantially enhances cell motility and is consistently associated with a poor clinical outcome. Finally, the mutually exclusive overexpression between AR, HOXC6 and NKX2-2 is preserved across various tissues and cancers, suggesting that mutual exclusivity may represent an intrinsic characteristic of driver TFs during tumorigenesis.

[1] Laboratory of Systems Biology, Shanghai Advanced Research Institute, Chinese Academy of Sciences, No. 100 Haike Road, Zhangjiang Hi-Tech Park, Pudong, Shanghai 201210, China. [2] Shanghai Institute of Materia Medica, Chinese Academy of Sciences, 555 Zuchongzhi Road, Zhang Jiang Hi-Tech Park, Pudong, Shanghai 201203, China. [3] College of Life Sciences, University of Chinese Academy of Sciences, No. 19A Yuquan Road, Shijingshan District, Beijing 100049, China. [4] School of Life Science and Technology, ShanghaiTech University, 393 Middle Huaxia Road, Pudong, Shanghai 201210, China. [5] School of Life Sciences, Shanghai University, 99 Shangda Road, BaoShan District, Shanghai 200444, China. * These authors contributed equally to this work. Correspondence and requests for materials should be addressed to P.W. (email: wangpeng@sari.ac.cn).

The function of androgen receptor (AR) is essential for the progression of prostate cancer. Hence, androgen-deprivation therapies have been the standard method to treat metastatic prostate cancer[1]. Although those therapies are initially effective, advanced prostate cancer eventually becomes refractory to androgen-ablation therapies, a stage termed 'castration-resistant prostate cancer' (CRPC), primarily by upregulating the expression of *AR*[2]. Importantly, *AR* gene amplification and overexpression are observed in approximately one third of CRPC patients[3,4], and studies have suggested that other mechanisms, such as mutations in AR and alterations in the coregulators of *AR*, are involved[1,2,4]. Recently, glucocorticoid receptor (GR) has been identified to reactivate the *AR* transcriptome in the absence of *AR* in patients resistant to enzalutamide[1], an anti-androgen drug, suggesting that mutual exclusivity, which broadly characterizes cancer-driving aberrations in signalling pathways[5], also dictates driver TFs that reactivate the *AR* transcriptome in CRPC.

MicroRNAs (miRNAs) are critical regulators of metastasis and numerous miRNAs display aberrant expression patterns in CRPC[6,7]. Integrative analyses have suggested that, unlike mRNAs, the differential expression of miRNAs in cancer is mostly unrelated to copy number variations[8], which indicates that TFs are likely factors dysregulating the oncogenic miRNA expression in CRPC. Indeed, *AR* has been shown to dysregulate several metastatic miRNAs in prostate cancer[6,9]. However, mechanisms underlying the aberrant miRNA expression in CRPC, particularly in the absence of *AR* overexpression, have not been systematically investigated.

Here, we perform a computational prediction and identify a module of mutually exclusive transcriptional drivers, *AR*, *HOXC6* and *NKX2-2*, which co-dysregulate a core set of metastasis-associated miRNAs in prostate cancer. Molecular, cellular and high throughput assays demonstrate that the identified TFs dysregulate the same miRNAs regulating key metastatic pathways and cellular motility. Moreover, analyses of multiple independent data sets show that overexpression of the identified TFs is consistently associated with a poor clinical outcome and the observed mutual exclusivity is highly conserved. Collectively, our findings indicate that mutual exclusivity, which represents an intrinsic property of oncogenic aberrations in signalling pathways, may also broadly characterize transcriptional drivers of gene regulatory networks during tumorigenesis.

## Results

**An algorithm to identify mutually exclusive driver TFs.** Our mutually exclusive driver TF hypothesis suggests that while the aberrant expression of metastasis-associated miRNAs is consistently observed across all metastatic cancer samples, each of the driver TFs dysregulating metastasis-associated miRNAs is only overexpressed in a subset of metastatic cancer samples, and in a mutually exclusive fashion. To facilitate computational prediction, we formulated the following criteria to nominate candidate driver TFs. First, instead of a single miRNA, a driver TF should dysregulate the vast majority of miRNAs differentially expressed in metastatic prostate cancer; second, cancer samples over-expressing the driver TFs should form a mutually exclusive pattern such that each driver TF dysregulates metastatic miRNA expression in a distinct subset of cancer samples, and these sets of samples demonstrate statistically significant tendencies towards mutual exclusivity; and third, those driver TFs should command statistically significant overlap in their miRNA targets, thus dysregulate the same metastasis-associated miRNA programme.

Our computational pipeline consisted of three major steps (Fig. 1). We first discretized the continuous gene expression

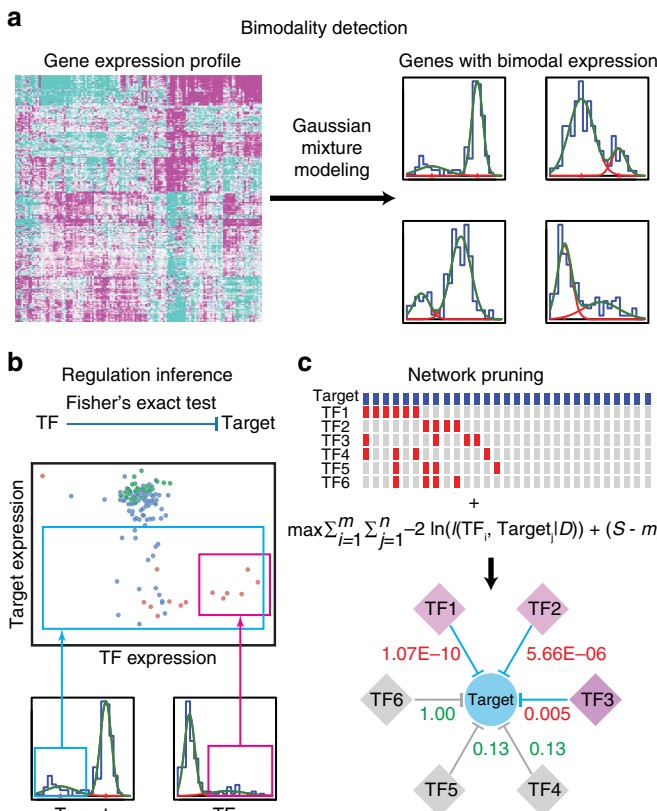

**Figure 1 | Overview of the computational approach to identify master TFs regulating microRNAs.** (**a**) Gaussian mixture modelling to identify bimodal expression patterns. (**b**) Inference of regulatory relationships between TF–gene pairs with Fisher's exact tests. (**c**) Network pruning with the MDL principle.

profiles of mRNA and miRNA into binary representations, designing the gene expression outliers enriched with metastatic samples as 'aberrations' (Fig. 1a). We then inferred the regulatory relationship between TFs and miRNAs by testing for significant associations between their aberrations using Fisher's exact tests, based on the observation that the transformation of primary cancers into metastatic ones represents a change of cellular state that is accompanied by aberrant activities of master drive TFs and consequently, altered expression of their target miRNAs (Fig. 1b). By using Fisher's exact tests over other statistical methods, such as correlation or mutual information, regulatory relationships can be effectively inferred with fewer samples. Moreover, Fisher's exact tests also capture TF–miRNA pairs that are un-correlated in normal samples, but gain aberrant regulations owing to the overexpression of driver TFs. To infer the driver TF-associated miRNA regulatory network, we employed a minimum description length (MDL) procedure (Fig. 1c), which seeks to explain the observed aberrant expression pattern of miRNAs (observed in all metastatic cancers) by a minimum number of aberrantly expressed TFs (each occurring in a subset of metastatic cancers). The MDL approach allows effective identification of driver TFs, which dysregulate more miRNAs than non-driver TFs, and the mutual exclusivity among driver TFs, which explains more observed aberrations than the same number of co-overexpressing TFs.

**Identifying candidate drivers in metastatic prostate cancer.** We applied our computational pipeline to infer mutually exclusive master TFs dysregulating metastatic miRNA expression using a

comprehensive cohort of prostate cancer samples (the Taylor data set), which includes mRNA and miRNA profiling for normal tissues, primary and metastatic prostate cancers[4]. An important assumption of our computational pipeline is that the abundance of an mRNA is a reliable indicator of the transcriptional activity of the corresponding TF. Although proteomics data, which is generally not available for large cohorts of tumour samples, provides a better approximation for the underlying transcriptional activity, the mRNA abundance is a reasonable indicator because recent genome-scale analyses demonstrate strong correlations between mRNAs and corresponding proteins[10]. The master TF regulatory network reconstructed by our method consisted of three master TFs demonstrating mutually exclusive overexpression, AR, HOXC6 and NKX2-2, and included nearly 20% (46 out of 233) of all examined miRNAs (Fig. 2a). Importantly, the reconstructed network suggested that the three TFs predominantly repress miRNA expression in metastatic prostate cancers because the vast majority (88%, 65 out of 74) of predicted regulations are inhibitions. As expected, the module was strongly associated with metastatic prostate cancer

because all but one of the tumour samples overexpressing the three master TFs were metastatic, and this group of samples captured nearly 80% (11 out of 14) of all metastatic samples (Supplementary Fig. 1). Moreover, unbiased clustering using expression profiles of all miRNAs and the three TFs identified a metastatic cluster containing 13 out of 14 metastatic prostate cancer samples. The metastatic cluster also included all samples overexpressing the three TFs (Supplementary Fig. 2A), further confirming the association between the AR–HOXC6–NKX2-2 module with metastatic prostate cancer. Strikingly, the predicted miRNA targets of the three master TFs displayed significant overlap: more than 75% of HOXC6 and NKX2-2 targets were also predicted to be AR targets (Supplementary Fig. 2B) ($P < 0.001$, Fisher's exact test), suggesting that AR, HOXC6 and NKX2-2 co-regulate a common set of miRNAs in a mutually exclusive fashion.

**The driver TFs dysregulate majority of metastatic miRNAs.** To experimentally validate the predicted results, we stably

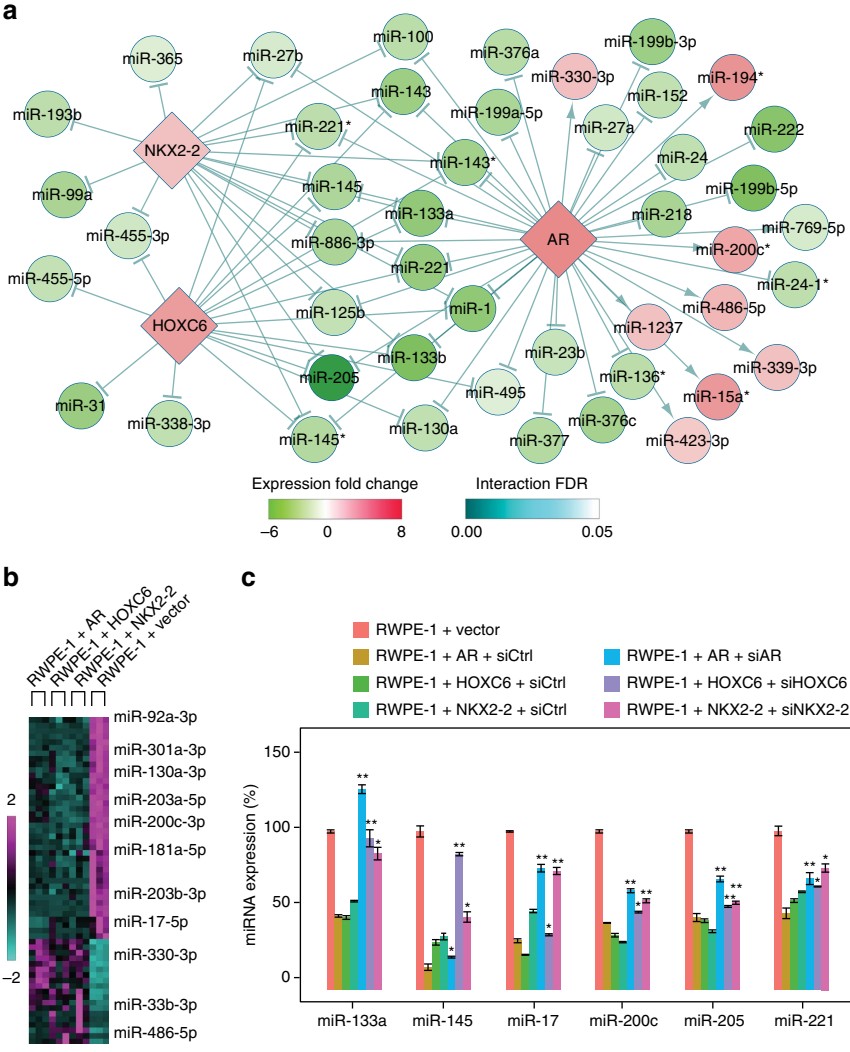

**Figure 2 | Computational identification of mutually exclusive TFs regulating a common set of microRNAs in metastatic prostate cancer. (a)** The microRNA gene regulatory network associated with the three TFs predicted by our computational pipeline. Circles represent microRNAs and diamonds represent TFs. Node colours represent expression changes. **(b)** Heatmap of miRNA-seq results showing differentially expressed microRNAs co-regulated by AR, HOXC6 and NKX2-2. **(c)** qRT-PCR results showing that the levels of target microRNAs were significantly elevated after the siRNA mediated silencing of AR, HOXC6 or NKX2-2. n = 3; error bars indicate mean ± s.d. *P < 0.05; **P < 0.01, as determined using the two-tailed Student's t-test. Results shown are representative of three independent experiments.

overexpressed *AR*, *HOXC6*, *NKX2-2* or a control empty vector in the benign prostate cell line RWPE-1 and analysed miRNA expression changes with deep sequencing (miRNA-seq) (Fig. 2b). We utilized the benign RWPE-1 cell line because the down-regulation of metastasis-associated miRNAs have already occurred in prostate cancer cell lines[4,11,12]. Hence, utilizing prostate cancer cell lines precludes the analyses of the suppressive effects of overexpressing the three TFs on metastasis-associated miRNAs, which is the key role of the three TFs as shown in the computationally reconstructed TF–miRNA regulatory network (Fig. 2a). Differential expression analysis identified a large number of miRNAs that were differentially expressed in cells expressing one of the master TFs: 226 for *AR*, 94 for *HOXC6* and 129 for *NKX2-2*. Consistent with our computational prediction, the miRNA-seq data showed that a significant number (59) of miRNAs were co-regulated by all three TFs ($P < 0.001$, Fisher's exact test), and majority (68%) of the co-regulated miRNAs were downregulated (Fig. 2b, Supplementary Fig. 2C). To further validate the predicted regulations, we measured the expression of selected miRNAs with quantitative real-time PCR (qRT-PCR), focusing on miRNAs with established roles in prostate cancer metastasis and computational predicted targets that failed to pass the false discovery rate (FDR) cutoff but demonstrated a consistent change of expression in the miR-seq analysis. Reassuringly, qPCR analysis validated the vast majority (22 out of 27) of tested TF–miRNA regulations (Fig. 2c, Supplementary Fig. 3A–D, Supplementary Table 1). Moreover, knocking down the expression of *AR*, *HOXC6* or *NKX2-2* with specific siRNAs abolished the expression changes induced by the three TFs (Fig. 2c, Supplementary Fig. 3A–D), suggesting that the regulations were likely to be direct. Overall, we were able to experimentally validate 18 (51%) of computationally predicted miRNA targets for *AR*, 7 (54%) predicted miRNA targets for *HOXC6* and 11 (73%) predicted miRNA targets for *NKX2-2* (Supplementary Table 1, Supplementary Fig. 3E). The average validation rate of our method is higher than 50%, which is comparable with reported validation rates of state-of-the-art network reconstruction algorithms[13] and significantly higher than random predictors (validation rates < 13%, see Methods). Taken together, qRT-PCR and miR-seq analyses demonstrated that *AR*, *HOXC6* and *NKX2-2* co-regulate 54 miRNAs (Supplementary Table 2).

To gain support of predicted regulations from an independent source, we queried published ChIP-seq data of *AR*, *HOXC6* and *NKX2-2* for peaks associated with miRNAs. Visualizing the binding events near selected miRNAs revealed that although *AR*, *HOXC6* and *NKX2-2* bound to distinct genomic locations, which was expected because they belong to different TF families, ChIP-seq peaks of all three TFs were readily observed near their validated miRNA targets, suggesting that they co-regulate these miRNAs (Supplementary Fig. 4A–C). Unlike *AR*, whose binding landscape has been documented by ChIP-seq analyses in prostate cancer cell lines[14], ChIP-seq data for *HOXC6* and *NKX2-2* are only available in LoVo cell[15], which is a colon adenocarcinoma cancer cell line. To robustly examine to what extend the three TFs co-regulate miRNAs on genome-scale, we performed bootstrapping analyses to mitigate the bias in these ChIP-seq data owing to the different epigenetic landscapes in these cell lines. Interestingly, while the three TFs' miRNA targets predicted from ChIP-seq data demonstrated significantly higher overlap than expected ($P < 0.01$, Fisher's exact test), no significantly higher overlap was observed near coding genes (Supplementary Fig. 4D,E). We then analysed mRNA expression changes induced by overexpressing the three TFs with microarray and confirmed that the three TFs indeed regulate distinct sets of coding genes (Supplementary Fig. 4F,G).

The central premise of our study is that the set of miRNAs co-regulated by *AR*, *HOXC6* and *NKX2-2* captures the majority of miRNAs associated with metastatic prostate cancer. We first evaluated this hypothesis using the Taylor data set, which we used to build the computational model. Differential expression analysis identified 29 metastasis-associated miRNAs (eight were upregulated and 21 were downregulated in metastatic prostate cancer, FDR < 0.01, absolute fold change ≥ 2, Supplementary Table 2). As expected, both the profile (using all metastasis-associated miRNAs) and individual metastasis-associated miRNAs (except for hsa-miR-548c-3p) demonstrated significant prognostic values (Supplementary Fig. 5). Critically, the set of miRNAs commonly dysregulated by *AR*, *HOXC6* and *NKX2-2* captured a significant percentage (12 out of 29, $P < 0.001$, Fisher's exact test) of the metastasis-associated miRNAs in the Taylor data set (Fig. 3a, Supplementary Table 2). The overlap was especially prominent for the six miRNAs that displayed large expression changes (absolute fold change > 4); in that five out of the six miRNAs (miR-1, miR-133a, miR-143, miR-145 and miR-205) were also co-regulated by the three TFs. To gain further support, we cross-validated the result using an independent data set reported by Hart *et al.*[16]. Similar to the Taylor set, the miRNAs co-regulated by *AR*, *HOXC6* and *NKX2-2* captured a large portion (15 out of 39, $P < 0.001$, Fisher's exact test) of metastasis-associated miRNAs in the Hart data set (Fig. 3a, Supplementary Table 2).

Interestingly, the Taylor set and the Hart set only displayed an insignificant overlap (five miRNAs) in miRNAs differentially expressed in metastatic prostate cancer (Fig. 3a, Supplementary Table 2), suggesting substantial bias in metastasis-associated miRNAs derived from genome-scale analyses. Hence, we queried pubmed for publications that provide experimental evidence of miRNAs associated with prostate cancer metastasis (Supplementary Table 2). Reassuringly, the vast majority (45 out of 54, 83%) of the miRNAs co-regulated by the three TFs have experimental evidence demonstrating their association with metastasis, and most of those miRNAs (30 out of 54, 56%) have been associated with prostate cancer metastasis. Finally, majority of the metastasis-associated miRNAs (35 out of 45) demonstrated a remarkable consistency in their reported regulatory roles in metastasis (promote or inhibit metastasis) and the changes of expression (downregulated or upregulated) induced by the three TFs, suggesting that the three TFs not only captured the membership of miRNAs associated with prostate cancer metastasis but also correctly demonstrated their functional roles in regulating metastasis.

**Functional validation of the mutually exclusive driver TFs.** While *AR* is a well-documented driver for prostate cancer, the roles of *HOXC6* and *NKX2-2* in the development of prostate cancer are poorly understood[17,18]. Although the overexpression of *AR*, *HOXC6* and *NKX2-2* significantly enhanced RWPE-1 cells' ability to proliferate, those cells failed to form colonies in soft agar assays (Supplementary Fig. 6A,B). The lack of transformation is consistent with our prediction that the three TFs are critical regulators of advanced stage, metastatic miRNAs, and previous reports showing that the overexpression of *AR* alone is not sufficient to form tumours[19]. Importantly, the substantial overlap between miRNAs co-regulated by the three TFs and miRNAs differentially expressed in metastatic prostate cancer suggested that the three TFs could play important roles in prostate cancer metastasis (Fig. 3a). Indeed, the miRNAs co-repressed by the three TFs include members of the miR-200 family, the miR-17-92 cluster and the miR-99a/let-7c/miR-125b-2 family (Fig. 2b, Supplementary Table 2). Those miRNAs are reported targets of *AR* and their downregulation has been shown to play an

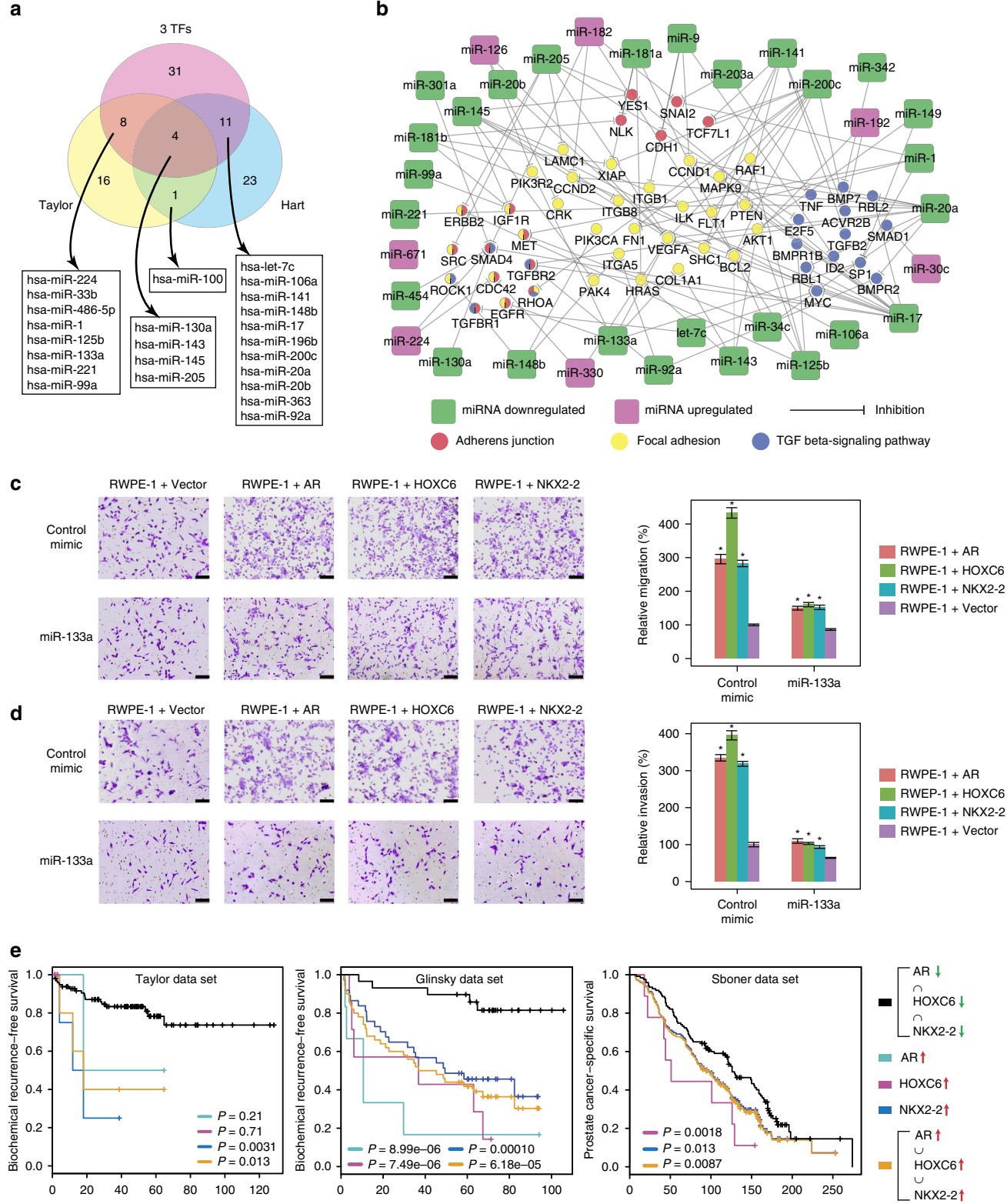

**Figure 3 | Functional and clinical validation of identified driver TFs.** (**a**) Venn diagram showing the overlap between experimentally validated microRNAs co-regulated by *AR*, *HOXC6* and *NKX2-2* and microRNAs differentially expressed in metastatic prostate cancers derived from two independent data sets. (**b**) The set of microRNAs co-regulated by *AR*, *HOXC6* and *NKX2-2* target genes in key metastatic pathways. Circles represent genes and squares represent microRNAs. The node colours represented the annotated function of the gene (for circles) and the expression change of the microRNA (for squares). (**c**) Representative images showing the results of the migration assay. Pictures in the top row are RWPE-1 cells overexpressing *AR*, *HOXC6*, *NKX2-2* or a control empty vector treated with control microRNA mimics. Pictures in the bottom row are cells treated with miR-133a mimics. Scale bars, 100 μm. The number of migrated cells were quantified and illustrated in the bar chart. *$P < 0.01$, as determined using the two-tailed Student's *t*-test. (**d**) Same as **c** for the invasion assay. (**e**) Kaplan–Meier survival analyses based on the overexpression of *AR*, *HOXC6* or *NKX2-2* in three independent prostate cancer data sets. Results shown are representative of three independent experiments.

important role in prostate cancer metastasis[6,9], suggesting that AR, HOXC6 and NKX2-2 promote prostate cancer metastasis via the downregulation of a common set of miRNAs. To validate this hypothesis, we first analysed whether those miRNAs co-regulated by AR, HOXC6 and NKX2-2 selectively target genes in key metastatic pathways. To this aim, we first complied a list of experimentally confirmed genes regulated by those miRNAs. We then analysed pathways enriched with genes regulated by those miRNAs and found that all significantly enriched pathways (FDR < 0.05) were associated with cancers or hallmark oncogenic pathways (Supplementary Table 3). Importantly, several key pathways associated with epithelial-to-mesenchymal transition (EMT) and metastasis, including TGF-beta signalling pathway, focal adhesion and adherent junction, were among the top enriched pathways, suggesting that those miRNAs could regulate EMT and metastasis (Fig. 3b, Supplementary Table 3).

We next examined the impact of AR, HOXC6 or NKX2-2 on RWPE-1 cell's motility by transwell assay. Consistent with pathway enrichment results, RWPE-1 cells overexpressing AR, HOXC6 or NKX2-2 demonstrated a significantly enhanced ability to migrate and invade comparing to RWPE-1 cells overexpressing a control empty vector (Fig. 3c,d, Supplementary Fig. 6C,D). Importantly, the increases in cell motility can be significantly reduced by mimics of miRNAs co-repressed by AR, HOXC6 and NKX2-2 but not by negative control mimics (Fig. 3c,d, Supplementary Fig. 6C,D), confirming that the impact of the three TFs on cellular motility is mediated by the repression of their target miRNAs. Finally, we analysed whether the overexpression of the three master TFs was associated with a poor clinical outcome by examining three independent prostate data sets. Reassuringly, patients overexpressing AR, HOXC6 or NKX2-2 were consistently associated with significantly worse clinical outcomes (Fig. 3e), suggesting that those driver TFs could provide a robust stratification of patients into clinically meaningful subtypes.

**The mutual exclusivity among driver TFs is highly conserved.** Next, We examined whether the observed mutual exclusivity between AR, HOXC6 and NKX2-2 overexpression is specifically associated with prostate cancer, or represents an intrinsic property of tumorigenesis. We first extended the mutual exclusivity analyses to three independent prostate cancer data sets[20–22], providing that they are of substantial size (sample count > 100) and that the three TFs demonstrated significant bimodality (bimodality index > 1.1)[23]. Reassuringly, all three data sets displayed strong mutually exclusive overexpression between the three TFs (Fig. 4a, Supplementary Fig. 7A,B). Moreover, four out of five samples violated the mutual exclusivity displayed low RPKM in HOXC6 or NKX2-2, which are more than an order of magnitude lower than that of AR in the same sample. Though numerically overexpressed, the low RPKM indicated that HOXC6 and NKX2-2 are not biologically overexpressed in those samples violating mutual exclusivity with AR overexpression.

We next examined the mutually exclusive overexpression between the three TFs across a large set of normal tissues and cell lines using the FANTOM5 gene expression data[24]. We noted that samples overexpressing each of the three TFs displayed a strong mutual exclusivity over 800 different tissues and cell lines (Fig. 4b, Supplementary Table 4), which suggested that they are lineage-specific master TFs in non-cancer tissues and they were aberrantly overexpressed during the development of metastatic prostate cancer. We next analysed their overexpression across 22 TCGA cancer data sets. Surprisingly, in all the 22 different cancers, the same mutual exclusivity was observed (Fig. 4c, Supplementary Fig. 7C). Thus, the mutually exclusive overexpression of driver TFs, similar to the mutual exclusivity

of mutations in oncogenic signalling pathways, is conserved across diverse cancer types. Moreover, the preserved mutual exclusivity indicates that mechanisms dictate tissue-specific overexpression of lineage-specific master TFs may be operational in various cancers, which could represent an important constrain limiting the pool of co-expressing driver TFs. Finally, this conserved pattern suggests that those TFs may play important roles in tumorigenesis of multiple cancers. Indeed, Kaplan–Meier survival analyses revealed that the aberrant overexpression of the three TFs were of significant prognostic values in several cancers, suggesting that they may be drivers in cancers of non-prostate origins (Supplementary Fig. 7D).

**Discussion**
Despite the development of numerous AR-based therapies, patients with metastatic prostate cancer invariable succumb to the disease, presumably owing to acquired resistance[1,2]. The finding that AR, HOXC6 and NKX2-2 co-dysregulate miRNAs that are crucial for metastasis in a mutually exclusive fashion resembles the role of reactivated GR in mediating acquired resistance to AR-based therapies[1], suggesting that mutually exclusive overexpression of mechanically related driver TFs may be a common mechanism underlying tumorigenesis. Importantly, the mutual exclusivity among driver TFs could provide novel insights into reducing the tremendous heterogeneity of cancer by mechanistically grouping mutually exclusive driver TFs into one oncogenic module. Indeed, samples overexpressing the three TFs (AR, HOXC6 and NKX2-2) together accounted for up to 80% of metastatic prostate cancer samples in the Taylor data set, suggesting that the combined inhibition of AR, HOXC6 and NKX2-2 may represent a more effective strategy to develop prostate cancer therapies.

An alternative strategy for combinatorial therapy is to target common upstream or downstream co-factors that are critical for the oncogenic activities of the master TFs. Recent studies suggested that super enhancers could stimulate the aberrant overexpression of cancer driver TFs and small molecules disrupting super enhancers, such as BRD4 inhibitors, have demonstrated utilities as novel therapeutics agents in numerous cancers[25–29]. Unfortunately, examining H3K27ac ChIP-seq data in multiple prostate cancer cell lines suggested that AR, HOXC6 and NKX2-2 are not broadly associated with super enhancers (Supplementary Table 5). However, because epigenetic landscapes could be substantially different between cancer cell lines and tumour tissues[30], future studies profiling super enhancers in prostate cancer tissues may help to delineate the roles of super enhancers in driving the overexpression of the master TFs, and consequently, in the downregulation of metastasis-associated miRNAs in prostate cancer.

Common co-factors mediating the repressive effects of the master TFs represent potential downstream candidates for therapeutic intervention, providing that AR, HOXC6 and NKX2-2 indeed employ identical co-factors. LSD1, which is a histone modification enzyme, mediates AR-based transcriptional repression of mRNAs in prostate cancer[31]. However, knocking down LSD1 did not upregulate metastasis-associated miRNAs in either LnCAP cells or in RWPE-1 cells overexpressing AR, HOXC6 or NKX2-2 (Supplementary Fig. 8A–D), suggesting that other co-factors mediate the transcriptional repression of miRNAs in prostate cancer. Importantly, promoter methylation has been associated the repression of metastasis-associated miRNAs in prostate cancer[32,33]. Interestingly, simultaneously overexpressing AR, HOXC6 and NKX2-2 in RWPE-1 cells did not further inhibit metastasis-associated miRNAs than RWPE-1 cells overexpressing a single master TF (Supplementary Fig. 8E,F). The lack of synergy among AR, HOXC6 and NKX2-2 suggests that the

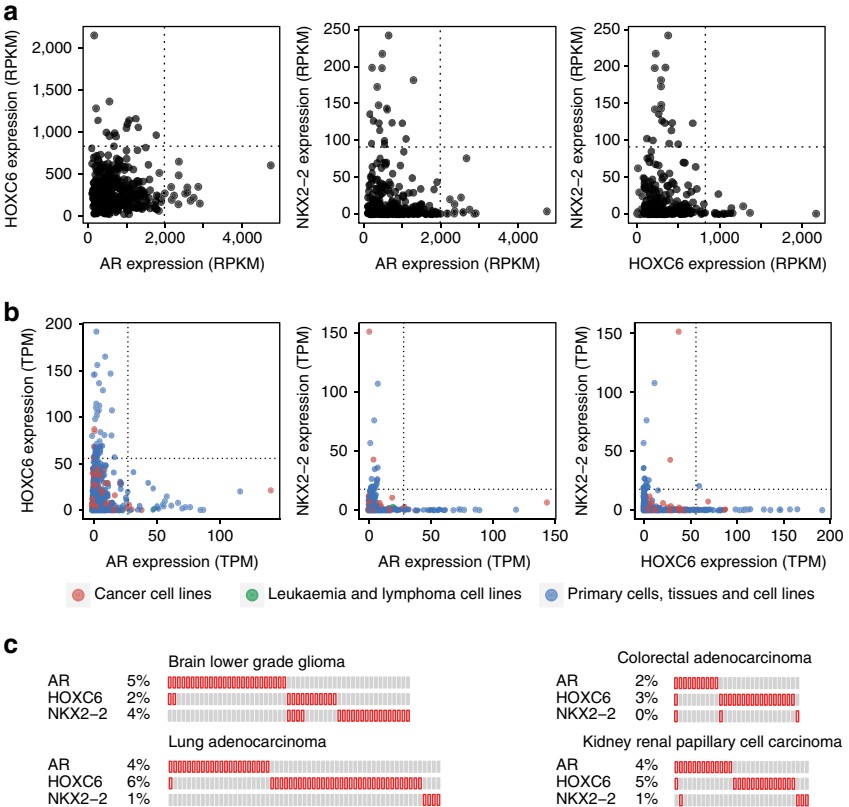

**Figure 4 | The mutually exclusive overexpression of driver TFs is preserved across various tissues and cancers. (a)** Scatter plots showing the mutually exclusive overexpression of *AR*, *HOXC6* and *NKX2-2* using TCGA prostate cancer gene expression data. Dotted lines represent cutoffs for overexpression. **(b)** Same as **a** using FANTOM5 gene expression data. **(c)** Oncoprints showing the mutually exclusive overexpression of *AR*, *HOXC6* and *NKX2-2* in representative TCGA cancers.

three master TFs may indeed compete for a common epigenetic co-factor to inhibit miRNA expression in prostate cancer. Future studies investigating the epigenetic mechanisms underlying the downregulation of metastasis-associated miRNAs in prostate cancer may help to resolve this issue.

The remarkable robustness of cellular functions is achieved through multilayered regulatory networks characterized by redundancy, feedback controls and modular design[34]. Consequently, oncogenic aberrations targeting distinct regulators of the same pathway could lead to the same cancerous outcome, which underlies the tremendous heterogeneity of cancer. While in theory there could exist many such oncogenic aberrations in a pathway, the finding that the identified mutual exclusivity among *AR*, *HOXC6* and *NKX2-2* was preserved across a diverse collection of cancers provided strong evidence that aberrations in TF-based drivers, similar to their counter parts in signalling pathways, are limited to a core set of drivers in a parsimony fashion. A potential mechanism underlying the mutually exclusive overexpression among *AR*, *HOXC6* and *NKX2-2* is that these TFs mutually inhibit the transcription of each other. However, simultaneously overexpressing *HOXC6* and *NKX2-2* in LnCAP cells did not inhibit *AR* expression (Supplementary Fig. 9). Moreover, the mRNA abundance of the three TFs does not demonstrate significant negative correlations (Supplementary Fig. 1). Taken together, these data suggest that mechanisms other than direct mutual inhibition dictate the mutually exclusive overexpression of the three TFs in metastatic prostate cancer. Master TFs are key attractors that integrate aberrant signals from multiple signalling pathways and manifest them via aberrant gene expression. Thus, future studies that integrate signalling pathways and gene regulatory networks may generate more comprehensive, clinically meaningful groupings

of cancer aberrations, which could represent an important step towards the ultimate goal of personalized medicine for cancer that each patient is associated with a set of actionable driver aberrations with effective therapies.

## Methods

**Identification of gene expression outliers.** We computationally reconstructed the TF–miRNA regulatory network using the Taylor data set[4]. We first identified gene expression outliers following the procedure proposed by cBioPortal[35] with important modifications. Instead of estimating normal sample's mean and variance using copy number neutral samples, which often demonstrate significant expression changes owing to transcriptional or epigenetic dysregulations, we analysed gene expression patterns using Gaussian mixture models and designed the cluster with largest number of samples as normal. We then calculated a $z$-score for each sample using the mean and standard deviation of the normal cluster and selected samples with an absolute $z$-score $> 2$ as true outliers. Finally, we analysed the overlap with metastatic samples in over- and under-expressed outliers and designated the one enriched with metastatic samples as aberrations.

**Pair-wise regulation inference with Fisher's exact tests.** To identify significant associations between TF-target pairs, we first generated a 'TF-based classification' of the target by calculating a new cutoff for target$_i$ based on the expression of TF$_j$. The new cutoff was defined as follows:

$$\text{cutoff}_{i,j} = \begin{cases} \max(\text{target}_{i,j}), & \text{if the target is under expressed} \\ \min(\text{target}_{i,j}), & \text{if the target is overexpressed} \end{cases}$$

where target$_{i,J}$ is the set of samples identified as the TF$_j$'s outliers.

A contingency table for the TF-target pair was then generated by comparing the sample classification based on the TF's cutoff and the sample classification based on cutoff$_{i,j}$ with the following modifications: (1) the true positive number was set to be the intersection of the TF's outliers and the target's outliers determined from the target's expression instead of cutoff$_{i,j}$ because the tested TF should not benefit from a 'relaxed' cutoff for the tested target in the association test. (2) The false positive number was set to be max(0, outlier.tf$_i$ − outlier.exp$_i$), where outlier.tf$_i$ was the number of outliers for target$_i$ determined by cutoff$_{i,j}$ and outlier.exp$_i$ was the

number of outliers for target$_i$ determined by the expression values of target$_i$. This modification removes the regulatory effects of 'hidden' TFs from the association test. Finally, significant regulations were identified using a FDR cutoff of 0.01.

**Inferring driver TF-associated miRNA regulatory network.** We then reconstructed the driver TF-associated miRNA regulatory network by applying the MDL principle. This was accomplished by maximizing the following objective function:

$$\max \sum_{i=1}^{m} \sum_{j=1}^{n} -2\ln\left(\mathcal{L}(\mathrm{TF}_i, \mathrm{Target}_j \mid D)\right) + (S - m)$$

where $S$ is the total number of candidate TFs, $m$ is the total number of TFs in the reconstructed network, $n$ is the number of targets and $\mathcal{L}(\mathrm{TF}_i, \mathrm{Target}_j \mid D)$ is the likelihood of a TF$_i$ regulating a Target$_j$ given the expression data $D$ determined by Fisher's exact tests.

**Identifying master TFs regulating a common set of miRNAs.** Master TFs were identified with binomial tests on the reconstructed TF–miRNA regulatory network using a $P$ value cutoff of 0.05. TFs sharing a significant number of targets were identified with Fisher's exact tests ($P < 0.01$). The mutual exclusivity among identified driver TFs was also checked with Fisher's exact tests and all three TFs demonstrated significant mutual exclusivity.

**Cell culture.** RWPE-1 and LnCAP cells were obtained from the Cell Bank of Chinese of Academy of Sciences. Cells were tested to be free of mycoplasma contamination and their identities were confirmed with STR DNA profiling. RWPE-1 cells were maintained in keratinocyte serum-free medium (Gibco) supplemented with $0.05\,\mathrm{mg\,ml}^{-1}$ bovine pituitary extract and $5\,\mathrm{ng\,ml}^{-1}$ human recombinant epidermal growth factor. LnCAP cells were maintained in RPMI-1640 (Gibco) supplemented with 10% FBS.

**RNA isolation and quantitative real-time PCR.** Total RNA was isolated from cells using TRIzol reagent (TaKaRa). Reverse transcription was performed using the PrimeScript RT Reagent Kit (TaKaRa) and gDNA Eraser (Perfect Real Time) or Fermentas K1622 RevertAid First Strand cDNA Synthesis Kit (Thermo Fisher Scientific) according to the manufacturers' instructions. qRT-PCR was performed using the SYBR Premix Ex Taq (Tli RNase H Plus) system on an ABI Step One Plus machine (Applied Biosystems). The experiments were performed in triplicate, and the values were normalized to that of U6 snRNA. The primer sequences were included in Supplementary Table 6.

**Transient siRNA knockdown and miRNA mimic transfection.** miRNA mimics and siRNAs were purchased from Shanghai GenePharma. AllStars negative mimic (Qiagen) or Silencer (Life Technologies) were used as negative controls. Cells were transfected with siRNA or micrRNA mimic at a final concentration of 100 pM using Lipofectamine 2000 reagent (Invitrogen) according to the manufacturer's instructions. Sequences for siRNAs and miRNA mimics are presented in Supplementary Table 7.

**Generation of RWPE-1 cell lines overexpressing driver TFs.** Sequences for AR (NM_000044), HOXC6 (NM_004503) and NKX2-2 (NM_002509) were synthesized by Genscript and cloned into the pLVX-IRES-Neo (Clontech) expression vector. Lentiviral stock preparation and cell infection were performed according to the manufacturer's instructions. Pools and clones of infected cells were maintained in the presence of Geneticin.

**Immunoblotting analyses.** Protein lysates were prepared in the presence of protease-inhibitor complex and phenylmethylsulphonyl fluoride. Twenty microgram aliquots were separated on 8% SDS–polyacrylamide electrophoresis gels, and the proteins were transferred onto a polyvinylidene difluoride membrane (Merck Millipore). The membrane was incubated for 1 h in blocking buffer (Tris-buffered saline containing 0.1% Tween (TBS-T), and 5% non-fat dry milk) followed by incubation overnight at 4 °C with the primary antibodies for AR (Abcam ab74272), HOXC6 (Santa Cruz sc-376330), NKX2-2 (Santa Cruz sc-15015) or LSD1 (Abcam ab37165). After washing with TBS-T, the blot was incubated with horseradish peroxidase (HRP)-conjugated secondary antibody, and the signals were visualized using an enhanced chemiluminescence system according to the manufacturer's instructions (Kodak). Alternatively, the gel images were analysed with Odyssey CLx fluorescence imaging system according to the manufacturer's instructions (LI-COR).

**Transwell migration and invasion assay.** The in vitro cell migration assay was performed using Transwell chambers (8-µm pore size; Costar). Cells were plated in serum-free medium ($2 \times 10^4$ cells per Transwell). Medium containing 10% FBS in the lower chamber served as a chemoattractant. After 48 h, the non-migrating cells were removed from the upper face of the filters using cotton swabs and the migratory cells located on the lower side of the chamber were stained with crystal violet, air dried, photographed and counted. Images of six random fields at $\times 10$

magnification were captured from each membrane, and the number of migratory cells was counted. Similar inserts coated with Matrigel were used to determine the cellular invasive potential in the invasion assay.

**MTS assay.** For MTS assay, cells were seed in 96-well plate with a density of $2 \times 10^3$ cells per well. To measure the absorbance, 20 µl of the CellTiter 96 AQueous One Solution Reagent (Promega) was added into each well and cells were incubated at 37 °C for 3 h. The absorbance was measured at 490 nm using a Microplate Reader (VersaMAx, Molecular Devices). All the experiment was repeated three times.

**Colony formation assay.** Cells were trypsinized into single-cell suspension for the colony formation assay. The bottom layer was prepared with a 0.8% agarose (Invitrogen) solution in culture medium in 6-well plates, and the gel was allowed to set for 20 min at room temperature. $5 \times 10^3$ cells were resuspended in 0.4% top agarose solution with culture medium (KSF-M medium with BPE for RWPE-1 cells; DEME medium with 10% FBS for A549 cells) and then were carefully placed on top of the bottom agarose in the six-well plates. The plates were incubated at 37 °C with 5% $CO_2$ and medium were replaced every 4 days. After two weeks, colonies were fixed with 1% formaldehyde and stained with 0.005% crystal violet in PBS for 1 h and counted under inverted microscope. Triplicate wells were measured in each treatment group.

**ChIP-seq and data analysis.** We used published ChIP-seq data from the Gene Expression Omnibus database for AR (GSE28264)[14], HOXC6 and NKX2-2 (GSE49402)[15]. The ChIP-seq peak calling and peak gene assignment were carried out using HOMER software (http://homer.salk.edu/homer/). Bootstrapping was carried out with R software by performing random sampling with replacement for 1,000 times.

**MicroRNA sequencing and microarray analysis.** The miRNA sequencing libraries were constructed according to the protocol for the Illumina small RNA Sample preparation kit. Sequencing was performed on the Illumina HiSeq 2,000 sequencer. Library construction and sequencing were performed at the Genergy Biotech (Shanghai). miRNA expression was analysed by miRdeep2.0.0.7 (ref. 36) and differentially expressed miRNAs were identified using an FDR cutoff value of 0.05. The mRNA expression profiling was carried out with Roche NimbleGen Human $12 \times 135$ K Gene Expression Array by KangChen Bio-tech. Raw data was processed with RMA algorithm and differential expression analysis was performed with R package limma[37] (Version 3.22.7).

**MicroRNA–gene regulation network.** Gene targets of miRNAs supported by strong experimental evidence (reporter assay or western blot) were downloaded from miRTarBase[38] (Version 4.5). The miRNA–gene interaction network was visualized by Cytoscape[39] (Version 2.8.3). GO enrichment analysis was performed by DAVID[40] (Version 6.7).

**Estimating the validation rate of a random predictor.** We employed a completely random predictor such that the predicted targets for a TF are all have 50% chance to be correct regardless of their rank in the list of predicted targets, which is equal to an AUC (Area Under receiver operating characteristic curve) of 0.5. To estimate the validation rate, we set the number of total miRNAs to be 1,800 (counting $-3p$ and $-5p$ as one as we did in the calculation of the validation rates for AR, HOXC6 and NKX2-2). We utilized the miR-seq data to estimate the number of true miRNAs targets to be 226, 94 and 129 for AR, HOXC6 and NKX2-2, respectively. We then calculated the validation rates by dividing the number of true miRNAs targets for each TF by the number of total miRNAs.

**Statistical and survival analysis.** Survival analysis was performed using the following public cancer data sets: the Taylor data set (GSE21032)[4], the Sboner data set (GSE16560)[41], the Glinsky data set[42] and TCGA data sets for 22 cancers. Normality of data and equal variance between different groups were confirmed before performing t-tests. Statistical tests and survival analyses were performed using R version 3.1.1 software.

**Code availability.** An R implementation of the algorithm to reconstruct TF–miRNA regulatory network can be accessed at https://github.com/mzxue/SIMBA.

**Data availability.** The data reported in this study have been deposited in GEO under accession number GSE71081 (miRNA-seq) and GSE81232 (mRNA array). The rest of the data that support the conclusions of this study are available from the corresponding author upon request.

### References

1. Arora, V. K. et al. Glucocorticoid receptor confers resistance to antiandrogens by bypassing androgen receptor blockade. Cell 155, 1309–1322 (2013).

# ARTICLE

2.  Chen, C. D. *et al.* Molecular determinants of resistance to antiandrogen therapy. *Nat. Med.* **10,** 33–39 (2004).
3.  Linja, M. J. *et al.* Amplification and overexpression of androgen receptor gene in hormone-refractory prostate cancer. *Cancer Res.* **61,** 3550–3555 (2001).
4.  Taylor, B. S. *et al.* Integrative genomic profiling of human prostate cancer. *Cancer Cell* **18,** 11–22 (2010).
5.  Ciriello, G., Cerami, E., Sander, C. & Schultz, N. Mutual exclusivity analysis identifies oncogenic network modules. *Genome Res.* **22,** 398–406 (2012).
6.  Mo, W. *et al.* Identification of novel AR-targeted microRNAs mediating androgen signalling through critical pathways to regulate cell viability in prostate cancer. *PLoS ONE* **8,** e56592 (2013).
7.  Watahiki, A. *et al.* MicroRNAs associated with metastatic prostate cancer. *PLoS ONE* **6,** e24950 (2011).
8.  Dvinge, H. *et al.* The shaping and functional consequences of the microRNA landscape in breast cancer. *Nature* **497,** 378–382 (2013).
9.  Shi, X. B. *et al.* An androgen-regulated miRNA suppresses Bak1 expression and induces androgen-independent growth of prostate cancer cells. *Proc. Natl Acad. Sci. USA* **104,** 19983–19988 (2007).
10. Edfors, F. *et al.* Gene-specific correlation of RNA and protein levels in human cells and tissues. *Mol. Syst. Biol.* **12,** 883 (2016).
11. Kobayashi, N. *et al.* Identification of miR-30d as a novel prognostic maker of prostate cancer. *Oncotarget* **3,** 1455–1471 (2012).
12. Boll, K. *et al.* MiR-130a, miR-203 and miR-205 jointly repress key oncogenic pathways and are downregulated in prostate carcinoma. *Oncogene* **32,** 277–285 (2013).
13. Margolin, A. A. *et al.* ARACNE: an algorithm for the reconstruction of gene regulatory networks in a mammalian cellular context. *BMC Bioinformatics* **7,** S7 (2006).
14. Tan, P. Y. *et al.* Integration of regulatory networks by NKX3-1 promotes androgen-dependent prostate cancer survival. *Mol. Cell. Biol.* **32,** 399–414 (2012).
15. Yan, J. *et al.* Transcription factor binding in human cells occurs in dense clusters formed around cohesin anchor sites. *Cell* **154,** 801–813 (2013).
16. Hart, M. *et al.* Comparative microRNA profiling of prostate carcinomas with increasing tumor stage by deep sequencing. *Mol. Cancer Res.* **12,** 250–263 (2014).
17. Ramachandran, S. *et al.* Loss of HOXC6 expression induces apoptosis in prostate cancer cells. *Oncogene* **24,** 188–198 (2005).
18. Khosravi, P. *et al.* Comparative analysis of prostate cancer gene regulatory networks via Hub type variation. *Avicenna J. Med. Biotechnol.* **7,** 8–15 (2015).
19. Stanbrough, M., Leav, I., Kwan, P. W., Bubley, G. J. & Balk, S. P. Prostatic intraepithelial neoplasia in mice expressing an androgen receptor transgene in prostate epithelium. *Proc. Natl Acad. Sci. USA* **98,** 10823–10828 (2001).
20. Beltran, H. *et al.* Divergent clonal evolution of castration-resistant neuroendocrine prostate cancer. *Nat. Med.* **22,** 298–305 (2016).
21. Robinson, D. *et al.* Integrative clinical genomics of advanced prostate cancer. *Cell* **161,** 1215–1228 (2015).
22. Cancer Genome Atlas Research N. The molecular taxonomy of primary prostate cancer. *Cell* **163,** 1011–1025 (2015).
23. Wang, J., Wen, S., Symmans, W. F., Pusztai, L. & Coombes, K. R. The bimodality index: a criterion for discovering and ranking bimodal signatures from cancer gene expression profiling data. *Cancer Inform.* **7,** 199–216 (2009).
24. The FANTOM Consortium and the RIKEN PMI and CLST (DGT). A promoter-level mammalian expression atlas. *Nature* **507,** 462–470 (2014).
25. Hnisz, D. *et al.* Super-enhancers in the control of cell identity and disease. *Cell* **155,** 934–947 (2013).
26. Loven, J. *et al.* Selective inhibition of tumor oncogenes by disruption of super-enhancers. *Cell* **153,** 320–334 (2013).
27. Whyte, W. A. *et al.* Master transcription factors and mediator establish super-enhancers at key cell identity genes. *Cell* **153,** 307–319 (2013).
28. Shu, S. *et al.* Response and resistance to BET bromodomain inhibitors in triple-negative breast cancer. *Nature* **529,** 413–417 (2016).
29. Asangani, I. A. *et al.* Therapeutic targeting of BET bromodomain proteins in castration-resistant prostate cancer. *Nature* **510,** 278–282 (2014).
30. Ziller, M. J. *et al.* Charting a dynamic DNA methylation landscape of the human genome. *Nature* **500,** 477–481 (2013).
31. Cai, C. *et al.* Androgen receptor gene expression in prostate cancer is directly suppressed by the androgen receptor through recruitment of lysine-specific demethylase 1. *Cancer Cell* **20,** 457–471 (2011).
32. Hulf, T. *et al.* Epigenetic-induced repression of microRNA-205 is associated with MED1 activation and a poorer prognosis in localized prostate cancer. *Oncogene* **32,** 2891–2899 (2013).
33. Formosa, A. *et al.* DNA methylation silences miR-132 in prostate cancer. *Oncogene* **32,** 127–134 (2013).
34. Stelling, J., Sauer, U., Szallasi, Z., Doyle, 3rd F. J. & Doyle, J. Robustness of cellular functions. *Cell* **118,** 675–685 (2004).
35. Gao, J. *et al.* Integrative analysis of complex cancer genomics and clinical profiles using the cBioPortal. *Sci. Signal.* **6,** pl1 (2013).
36. Friedlander, M. R. *et al.* Discovering microRNAs from deep sequencing data using miRDeep. *Nat. Biotechnol.* **26,** 407–415 (2008).
37. Ritchie, M. E. *et al.* Limma powers differential expression analyses for RNA-sequencing and microarray studies. *Nucleic Acids Res.* **43,** e47 (2015).
38. Hsu, S. D. *et al.* miRTarBase update 2014: an information resource for experimentally validated miRNA-target interactions. *Nucleic Acids Res.* **42,** D78–D85 (2014).
39. Shannon, P. *et al.* Cytoscape: a software environment for integrated models of biomolecular interaction networks. *Genome Res.* **13,** 2498–2504 (2003).
40. Huang, da, W., Sherman, B. T. & Lempicki, R. A. Systematic and integrative analysis of large gene lists using DAVID bioinformatics resources. *Nat. Protoc.* **4,** 44–57 (2009).
41. Sboner, A. *et al.* Molecular sampling of prostate cancer: a dilemma for predicting disease progression. *BMC Med. Genomics* **3,** 8 (2010).
42. Glinsky, G. V., Glinskii, A. B., Stephenson, A. J., Hoffman, R. M. & Gerald, W. L. Gene expression profiling predicts clinical outcome of prostate cancer. *J. Clin. Invest.* **113,** 913–923 (2004).

## Acknowledgements

The authors would like to thank Dr Ying Xu for critical comments on the manuscript and members of the Laboratory of Systems Biology for helpful discussions. This work was supported in part by the National Natural Science Foundation of China (NSFC) grant 31271413, Science and Technology Commission of Shanghai Municipality Project 16ZR1449000 and a startup fund from the Chinese Academy of Sciences 'Hundred Talent Program' to P.W.

## Author contributions

M.X. implemented the computational pipeline and performed computational analyses. H.L., L.Z. and H.C. performed the experiments and contributed to computational analyses. Y.L., S.D. and Y.Y. contributed to experiments. P.W. designed the algorithm, supervised the study and wrote the manuscript. All authors approved the final manuscript.

## Additional information

**Competing interests:** The authors declare no competing financial interests.

