## [Peer review file · Nature Communications]

Reviewers' comments:

Reviewer #1 (Remarks to the Author):

Summary: Integrative analyses in prostate cancer have suggested that, unlike mRNAs, the differential expression of microRNAs is mostly unrelated to copy-number variations (CNVs), which leads to the hypothesis that TFs are likely factors regulating microRNA expression. The authors performed a computational prediction followed by experimental validations and report that AR, HOXC6 and NKX2-2 are mutually exclusive driver TFs whose overexpression induces the aberrant expression of the microRNAs dysregulated in metastatic prostate cancer.

Major comments:

- The main theme of the manuscript is the “mutually exclusive relationship” between the master TFs AR, HOXC6 and NKX2-2 as regulators of expression of microRNAs in prostate Ca. While, in general, the mRNAs for these TFs are overexpressed (outliers) in different samples, there are (a few) samples with “overexpression” of more than one of these TFs in Fig. 4 and S1. Moving the cutoffs left/right or down/up would change (increase/decrease) the potential overlap for outliers. In the Methods, the authors describe that they “calculated a z-score for each sample using the mean and standard deviation of the normal cluster and selected samples with an absolute z-score > 2 as true outliers”, but that cut-off may not necessarily be the most biologically significant threshold. Thus, is the term “mutually exclusive” the best descriptor in this case (as it is a very absolute term, and, strictly speaking, inaccurate in this case), or would it be best to say that there is “strong tendency against co-expression” of these master TFs and variable dependency on them in various specimens?

- Could the authors please clarify which patient mRNA and microRNA datasets were used to build their core model/analysis (e.g. in Fig. S1 and S2)? It is not clear in the Methods. Are they the GSE71081 and GSE81232 (not publicly available yet)?

- The title of the manuscript is “Computational identification of mutually exclusive transcriptional drivers regulating metastatic microRNAs in prostate cancer”, giving the impression that these master TFs are driving = promoting the expression of these miRNAs. However, most of these microRNAs are actually suppressed by these TFs (Fig. 2A) but this is not clearly discussed. Instead, the non-committal term “regulate” is used throughout the manuscript, which leaves a lot of vagueness.

- In view of the reported suppressive effect of the master TFs AR, HOXC6 and NKX2-2 on expression of many microRNAs in prostate Ca, exactly how is this effect accomplished? In the case of AR, it has been reported that AR recruits Lysine Specific Demethylase 1 (LSD1/KDM1) to directly suppress gene expression (Cai C, et al. Cancer Cell. 2011;20(4):457-471). Is this a mechanism for these three master TFs in this case of suppressing microRNA expression?

- Why are these master TFs AR, HOXC6 and NKX2-2 “mutually exclusive” in their expression patterns? Do they suppress each other's expression?

- How do these master TFs AR, HOXC6 and NKX2-2 manage to regulate the expression of a common microRNA panel? Do they bind to the same DNA motif? Based on the ChIP-Seq data used in the manuscript, in each microRNA locus, do these three master TFs bind to the same location (in different specimens, obviously)? At least the examples shown in Fig. S4 do not appear to support this hypothesis.

- An important point to be made is that the expression of the three master TFs AR, HOXC6 and NKX2-2 is quantified in patient datasets at the mRNA level. Their protein levels could be substantially different from what the mRNA indicates, and the transcriptional activity even more different. For

example, active AR suppresses its own mRNA. So, it is far from ideal to study TF mRNA levels and extrapolate to their transcriptional activity.

- The authors provide the GEO numbers for the ChIP-Seq datasets used, but it would be helpful to discuss in the text which tissue/cell line and conditions/treatment were applied. For example, it appears that the HOXC6 and NKX2-2 ChIP-Seqs were performed in colorectal, not prostate cells. How are those datasets relevant to prostate cancer?

- The experimental validation rate of computationally predicted microRNA targets: 51% for AR, 54% for HOXC6 and 73% for NKX2-2 (Table S1, Fig. S3E) is rather low. A validation rate close to 50% is like flipping a coin.

Minor points:

- Throughout the manuscript/figures, wherever RWPE1 cells were used, the control condition is labelled "RWPE1" instead of "control vector" or "control siRNA" etc. However, the treated conditions are labeled with the actual treatment X, frequently without "RWPE1 + X", creating confusion as to what is treatment and what is the cell line model.

- Several typos are present throughout the manuscript, e.g. FigS1 "Mormal".

- In the Abstract: "Androgen-ablation therapies, which are the standard treatment for metastatic prostate cancer, invariably succumb to acquired resistance". The verb succumb usually refers to patients. For the therapies, in this context, I would suggest to use the term "lead to" or "are complicated by" or "are limited by the emergence of".

Reviewer #2 (Remarks to the Author):

The work is using computational inference and experimental approaches to find TF-miRNA modules. It is interesting and the methods are logical. I have minor issues: (1) For the Fisher' test, authors conducted multiple tests. Even the authors used 0.001 as a cutoff for p value, there are still lots of false-positives. A proper control is necessary. (2) For the modules , When taking looking the targets. Are all of them enriched with cancer hallmarks (PMID: 247476960). In general, the hypothesis (PMID: 24747696) believes that there are many ways to trigger cancer progression, but they will converge to cancer hallmarks.

Reviewer #3 (Remarks to the Author):

In this work Xue et al. predicted computationally that 3 transcription factors (TF): AR, HOXC6 and NKX2-2 are mutually exclusive drivers in metastatic prostate cancer, and their overexpression induces the aberrant expression of metastasis-associated miRNAs. Their results were also validated experimentally in a benign prostate cell line. The authors suggest that mutual exclusivity among driver TFs could provide novel insights into reducing the tremendous heterogeneity of cancer by mechanistically grouping mutually exclusive driver TFs into one oncogenic module.

Overall, this is a well-designed and carefully-executed research work, which I believe deserves to be published. Although the involvement of AR is widely-known in prostate cancer, the general idea of these 3 mutually exclusive TFs act as drivers for overexpressed miRNAs in the metastatic form of the disease, is novel. The results presented here, seem to be of importance to scientists in the specific field and the idea of mutually exclusive TFs as drivers for overexpressed miRNAs is quite interesting to researchers in other related disciplines.

Nevertheless, I believe that there are some concerns that need to be answered.

Main comments:

1) Overall, the number (n=3) of TFs found to be mutually exclusive drivers whose overexpression induces the aberrant expression of metastasis-associated miRNAs in metastatic prostate cancer, seems indeed rather small. I sense that the criteria followed during the computational identification of these mutually exclusive driver TFs, was rather strict and perhaps various inconsistencies detected between this study and various publicly available datasets, are due to these strict criteria applied.

2) Did the authors try to over-express or silence simultaneously all 3 TFs in RWPE-1 cell line?

3) ChIP-seq results represent just a small fraction of the number of miRNAs that are located around the peaks of all 3 TFs, while the authors report a large number of them. How far from the corresponding AR, HOXC6 and NKX2-2 binding sites were the rest of the miRNAs located?

4) The small number (n=4) of overlap in differentially expressed (DE) miRNAs among the Taylor & Hart datasets and the 3 TFs' targets, is rather alarming. A similar workflow of judging which miRNAs are differentially expressed should have been taken into careful consideration. Of course, the authors have solved this problem by querying Pubmed for publications that provide evidence that their miRNAs are associated with prostate cancer metastasis. However, this type of evidence was not detected for all of their miRNAs (but only for 56% of them), and this is worrying. Furthermore, I was wondering what would the overlap with the Glinsky and Sboner datasets be?

5) Is there any association between AR, HOXC6 and NKX2-2 and Super-Enhancer regions detected in the prostate cancer genome? Perhaps the authors could search this using publicly available ChIP-seq data.

Minor comments:

1) Perhaps if the authors validated more miRNAs as TF targets by qPCR, the validation rate would be higher than 50%. I think that the number (n=6) of miRNAs chosen for qPCR validation in page 7 is rather low, compared to the larger number of co-regulated miRNAs (n=54).

2) Perhaps it would be worthy if the cell culture-related experiments were repeated with another benign prostate cell line, apart from RWPE-1.

- 3) Line 149-15: "RWPE-1 was preferred over prostate cancer cell lines because the dysregulation of metastasis associated microRNAs have already occurred in those cancerous cell lines". Do the authors mean that dysregulation of metastasis associated microRNAs has been reported previously in this cell line (by the corresponding cited works)?
- 4) Fig S1: Please replace the words "MORMAL" and "Hsa-miR-205" with "NORMAL" and "hsa-miR-205", respectively (vertical axis) in the legend.
- 5) Line 140: "Fig. S2B-C" should be mentioned.
- 6) Line 141: "no significant overlap was observed near coding genes". Figure S4E represents the overlap between all genes (coding and non-coding) and is misleading with the above sentence. Judging from Figure S4E, it seems that there's a large number of overlapping near coding genes (1,631 genes). The authors could represent here the overlap between just the coding genes that are associated with AR, HOXC6 and NKX2-2 binding events derived via bootstrapping of published ChIP-seq data (and exclude the non-coding genes).
- 7) Lines 184-186: What microarrays were used for further validation of the mRNA expression changes induced by overexpressing AR, HOXC6 and NKX2-2? It is not mentioned in the Materials & Methods.
- 8) Line 176: Please cite the published ChIP-seq data used for bootstrapping analysis.
- 9) Line 191: Please cite the Taylor data set used.
- 10) Line 196: Please replace "has-miR-548c-3p" with "hsa-miR-548c-3p".
- 11) Line 417: Please replace "Has-miR-200c-3p" with "hsa-miR-200c-3p".
- 12) Line 488: "...cells were incubated at 37oC for 3 h."
- 13) Figure 4: Please revert the axes of the Scatter plot showing the mutually exclusive overexpression between HOXC6 and NKX2-2, accordingly, between subfigures A and B.
- 14) The role of epigenetic mechanisms affecting dysregulated AR gene expression could be mentioned in the discussion.

Reviewer #1.

Main comments:

1. The main theme of the manuscript is the “mutually exclusive relationship” between the master TFs AR, HOXC6 and NKX2-2 as regulators of expression of microRNAs in prostate Ca. While, in general, the mRNAs for these TFs are overexpressed (outliers) in different samples, there are (a few) samples with “overexpression” of more than one of these TFs in Fig. 4 and S1. Moving the cutoffs left/right or down/up would change (increase/decrease) the potential overlap for outliers. In the Methods, the authors describe that they “calculated a z-score for each sample using the mean and standard deviation of the normal cluster and selected samples with an absolute z-score > 2 as true outliers”, but that cut-off may not necessarily be the most biologically significant threshold. Thus, is the term “mutually exclusive” the best descriptor in this case (as it is a very absolute term, and, strictly speaking, inaccurate in this case), or would it be best to say that there is “strong tendency against co-expression” of these master TFs and variable dependency on them in various specimens?

We agree that the cutoff may not be the most biologically significant one for a particular set of data. However, it is not appropriate to move cutoffs. The detection of outliers, and mutual exclusivity based on outliers, are similar to the detection of differentially expressed genes in that though many biologically relevant genes maybe below the cutoff used, a sound statistical procedure (z-score in our case, p-value, FDR etc in differential expression analyses) and an appropriate cutoff (such as z-score > 2 or FDR < 0.01) should be applied. There are controversies associated with p-value, which is often misinterpreted, but there are currently no clear alternatives and certainly not by an ad hoc cutoff, which probably varies with different data sets, to capture more biologically significant samples.

The term “mutually exclusive” should be interpreted from a statistical perspective in that as far as two or more genes demonstrate statistically significant tendency towards mutual exclusivity, they should be considered as mutually exclusive. Indeed, mutual exclusivity analyses have been extensively utilized in the analysis of TCGA and other multi-dimensional cancer genomics data, and statistical test, rather than requiring NO sample overlap, has been the standard practice to determine mutual exclusivity.

Finally, it is common to have two well-established mutually exclusive genes to display overlap in their outlier samples. For example, TP53 and MDM2 are textbook mutually exclusive cancer driver genes, but there are numerous cancer samples simultaneously harboring TP53 mutation and overexpressing MDM2. Despite the numerous overlap, TP53 and MDM2 are considered to be mutually exclusive both from their biological roles in cancer and from statistical significance. As an example, we attached Fig. 4B of the TCGA breast cancer paper (doi:10.1038/nature11412) below, which shows multiple cancer patients simultaneously harboring TP53 mutation and MDM2 overexpression.

2. Could the authors please clarify which patient mRNA and microRNA datasets were used to build their core model/analysis (e.g. in Fig. S1 and S2)? It is not clear in the Methods. Are they the GSE71081 and GSE81232 (not publicly available yet)?

The dataset used is the Taylor data set, which has been described in results section (first paragraph of page 6), where we elaborated why this data set is appropriate for our study. We agree that description of data sets should be put into the method section to make it clear which data set is used for the algorithm. To clarify this issue, we have updated the manuscript by clearly stating that the Taylor data set was used in the algorithm both in the results and methods sections. GSE71081 and GSE81232 are the miRNA-seq and mRNA array we generated in this study to validate the roles of AR, HOXC6 and NKX2-2 on miRNA and mRNA expression.

3. The title of the manuscript is “Computational identification of mutually exclusive transcriptional drivers regulating metastatic microRNAs in prostate cancer”, giving the impression that these master TFs are driving/promoting the expression of these miRNAs. However, most of these microRNAs are actually suppressed by these TFs (Fig. 2A) but this is not clearly discussed. Instead, the non-committal term “regulate” is used throughout the manuscript, which leaves a lot of vagueness.

We thank reviewer for pointing out this issue, which is very important for the scientific accuracy of our manuscript. We agree that the dominant role of the TFs is to repress miRNA expression in metastatic prostate cancer, which is consistent with the role of the key miRNAs suppressed by the three TFs, such as members of the miR-200 family, in inhibiting metastasis. We have modified the manuscript and clearly discussed that the dominant role of the TFs is to repressed key miRNAs to promote metastasis. Specifically, new discussions were added in the abstract, the results pertaining the computationally predicted TF-miRNA network, miRNA-seq results and the transwell functional assays. We also replaced “regulate” with “dysregulate” throughout the manuscript to give the impression that those TFs drive the abnormal expression of metastasis-associated miRNAs.

4. In view of the reported suppressive effect of the master TFs AR, HOXC6 and NKX2-2 on expression of many microRNAs in prostate Ca, exactly how is this effect accomplished? In the case of AR, it has been reported that AR recruits Lysine Specific Demethylase 1 (LSD1/KDM1) to directly suppress gene expression (Cai C, et al. Cancer Cell. 2011;20(4):457-471). Is this a mechanism for these three master TFs in this case of suppressing microRNA expression?

We agree that this is a very interesting mechanism to explore and have carried out new experiments to examine the role of LSD1 in repressing miRNAs in prostate cancer. We knocked down LSD1 with specific siRNAs and measured the impact on the expression of six miRNAs (has-miR-133a, has-miR-145, has-miR-17, has-miR-200c, has-miR-205 and has-miR-221) in RWPE-1 cells overexpressing AR, HOXC6 or NKX2-2. However, we didn't observe significant upregulations in the expression of tested miRNAs. To further validate the results from RWPE-1 cells, we also knocked down LSD1 in LnCAP cells. Consistent with the results from RWPE-1 cells, knockdown LSD1 didn't significantly enhance the expression of the six miRNAs in LnCAP cells. Results of those experiments have been added into the manuscript as supplementary Figure S8A-D.

Studies have shown that hypermethylation at promoters as a common mechanism of miRNA repression in metastatic prostate cancer (Hulf et al., *Oncogene*. 2013 Jun 6;32(23):2891-9 and Formosa et al., *Oncogene*. 2013 Jan 3;32(1):127-34). LSD1 has been reported to suppress gene expression by inducing histone demethylation (Cai C, et al. *Cancer Cell*. 2011;20(4):457-471). Thus, it appears that LSD1, which is a histone modification enzyme, may not directly suppress miRNAs expression by inducing promoter methylation. However, studies have shown that histone modification and promoter methylation are interconnected processes (Rothbart et al., *Nature structural & molecular biology*, 19 (11), 1155-60). Hence, it is possible that other histone modification enzymes maybe involved in the repression and promoter methylation of metastatic miRNAs in prostate cancer. Future studies analyzing the epigenetic mechanisms regulating miRNA repression in prostate cancer may help to resolve this issue.

We have added a paragraph in discussion to discuss the results of LSD1 related experiments and the epigenetic mechanisms inhibiting miRNA expression in prostate cancer, which is also suggested by reviewer #3.

5. Why are these master TFs AR, HOXC6 and NKX2-2 “mutually exclusive” in their expression patterns? Do they suppress each other’s expression?

While the TFs repressing each other is an attractive mechanism, it is unlikely because their gene expression levels don't display significant negative correlations (Fig. S1). Moreover, we performed new experiments by simultaneously overexpressing HOXC6 and NKX2-2 in LnCAP cells, and didn't observe significant changes in AR expression, which suggests that the TFs don't directly repress each other (results has been added into the manuscript as Supplementary Figure S9).

The mutual repression theory is analogous to synthetic lethal where two incompatible mutations could induce cell death, thus lead to mutual exclusivity. However, synthetic lethal is probably not the general mechanism behind observed mutual exclusivity in cancer pathways or in the AR-HOXC6-NKX2-2 modules, because individual tumor sample violating the mutual exclusivity rule is commonly observed (as shown in our respond to comment #1)

We consider a random mutation process during tumor evolution, which is a simpler mechanism, to be a more probable mechanism to generate the mutually exclusive pattern in TFs and other cancer-related genes.

During the development of cancer, progenitor cancer cells randomly acquire mutations. Once a driver, such as copy number amplification of AR, is acquired, the cell will gain considerable growth advantage and expand. Because mutations occur randomly in human genome at low rates, a second mutation leads to overexpression of HOXC6 or NKX2-2 will have a low rate of occurrence in AR-high cancer cells. Conversely, AR mutation will also have a low occurrence in HOXC6-high or NKX2-2 high cells. Hence, the low rate of mutation and the large size of human genome could naturally lead to mutually exclusive patterns.

It is important to note that cancer drivers could have synergistic effects, which will lead to co-occurrence of those drivers, because their combined effects will give rise to cells with greater advantage to outgrow other clones. However, such synergy is unlikely to occur between AR, HOXC6 and NKX2-2. Because those master TFs largely repress the same set of miRNAs. Hence, acquiring the overexpression of a second TF will not gain further advantage. During revision, we simultaneously overexpressed the three TFs (as suggested by reviewer #3) in RWPE-1 cells and didn't observe a significantly increased suppression of target miRNAs, which further suggests that the three TFs suppress their target miRNAs independently (Fig. S8E-F).

The argument that mutual exclusivity is a consequence of random mutation in cancer can also help to explain the mutual exclusivity among molecules in signaling pathways. The expression of molecules in the same signaling pathway is often correlated, or transcriptionally co-regulates as a module (Segal et al., Nature Genetics 37, S38 - S45 (2005)). Hence, random mutation, rather than mutual repression, could be a more probable explanation.

Finally this issue deserves a new project to thorough examine various mechanisms, including the mutual repression suggested by the reviewer, in generating the mutual exclusivity patterns among cancer genes, and we appreciate reviewers for raising this insightful point.

We have added a new section in discussion to briefly discuss this issue together with the new results from overexpressing both HOXC6 and NKX2-2 in LnCAP cells.

6. How do these master TFs AR, HOXC6 and NKX2-2 manage to regulate the expression of a common microRNA panel? Do they bind to the same DNA motif? Based on the ChIP-Seq data used in the manuscript, in each microRNA locus, do these three master TFs bind to the same location (in different specimens, obviously)? At least the examples shown in Fig. S4 do not appear to support this hypothesis.

This is a misunderstanding that we would like to clarify. It is important to note that distinct TFs could

regulate the same gene while binding to different locations in the genome, as far as their binding sites are near the same target genes, or as shown by numerous 3-D chromosome conformation studies, could loop to the same target genes. They don't bind to the same motif, which is expected because they belong to different TF families. They also don't bind to the same location as noted by the reviewer (as shown in Fig. S4A-C). However, they do bind to nearby locations of the same gene as shown in Figure S4A-C, which indicates that they could regulate same genes.

As outlined in the manuscript, we assigned ChIP-seq peaks to genes using the standard procedure in the field, and assumed that a TF regulates a miRNA if ChIP-seq peaks of the TF are assigned to the miRNA. We then performed Fisher's exact tests to determine whether the miRNA targets predicted from ChIP-seq data demonstrate significantly higher overlap than random. Importantly, statistically analysis did demonstrate that the three TFs have a significantly higher tendency to regulate same miRNAs than expected using the ChIP-seq data (Fig. S4D).

The misunderstanding is largely a result of our presentation. We have elaborated the procedure in the manuscript, which should readily clarify this issue.

7. An important point to be made is that the expression of the three master TFs AR, HOXC6 and NKX2-2 is quantified in patient datasets at the mRNA level. Their protein levels could be substantially different from what the mRNA indicates, and the transcriptional activity even more different. For example, active AR suppresses its own mRNA. So, it is far from ideal to study TF mRNA levels and extrapolate to their transcriptional activity.

We agree that protein abundance is a better indicator of the corresponding transcriptional activities. Unfortunately, large-scale proteomics, or other types of data measuring protein levels in large cohorts of cancer samples are rather rare. While TCGA is making progress, the data for prostate cancer is not available yet.

Given the lack of proteomics data, the only rational approach is to extrapolate protein abundance from mRNA expression data, which has been a common practice in the large body of literatures attempting to utilize the tremendous amount of microarray or rna-seq data to identify putative cancer drivers.

Many studies have analyzed the correlation between the abundance of mRNA and protein and have generated contradicting results. Importantly, a recently study (Gene-specific correlation of RNA and protein levels in human cells and tissues, Edfors et al., *Molecular systems biology* (2016) 12, 883) has resolved the issue and shown that "protein levels correlate well with transcript levels, if a gene-specific and cell/tissue-independent RNA-to-protein (RTP) conversion factor is introduced". The study demonstrated that mRNA abundance is highly correlated with and is the main factor to determine protein abundance. Thus when comparing the same gene, such as AR, across different samples, we can be reasonably certain that a sample expressing higher level of AR mRNA is also expressing higher level of AR proteins, and consequently, displaying higher AR transcriptional activity.

AR regulates its own expression is a key mechanism for the homeostasis of AR expression. Because the overexpression of AR mRNA and protein in prostate cancer tissues and cell lines are well documented, such auto-regulation doesn't prevent AR from being overexpressed, it mainly provides a key mechanism to prevent AR expression from going over the roof, which probably be detrimental even to cancer cells.

Moreover, the key extrapolation that overexpressed master TFs repress miRNAs has been experimental validated in our study. Taken together, we can be reasonably confident in the transcriptional activity extrapolated from mRNA expression.

Finally, we totally agree that the lack of protein expression is often overlooked in studies utilizing gene expression data. We thank reviewer for pointing out this important issue and have discussed the limitation and rationale in extrapolating transcriptional activity from mRNA expression data when the Taylor data set was first introduced in the result section.

8. The authors provide the GEO numbers for the ChIP-Seq datasets used, but it would be helpful to discuss in the text which tissue/cell line and conditions/treatment were applied. For example, it appears that the HOXC6 and NKX2-2 ChIP-Seqs were performed in colorectal, not prostate cells. How are those datasets relevant to prostate cancer?

This is a very insightful point. We have given this issue a thorough consideration before we utilized those data and we will elaborate our reasons below.

Our original goal to utilize published ChIP-seq data is to provide independent evidence that the 3 TFs directly regulate their microRNA targets. Because the binding of a TF to genomic regions is mainly determined by the structure of the TF and the target DNA sequences, which are both large invariant across cell lines, the ChIP-seq data from different cell lines could largely serve this purpose.

Comparing ChIP-seq data from the same cell line, ChIP-seq data generated from different cell lines will suffer additional bias owing to the difference in the epigenetic landscapes, which will impact the accessibility of TFs to their binding sites. Though ChIP-seq data from the same cell line is preferred, ChIP-seq from same cell line also have limitations. Studies from ENCODE have shown that for most analyzed TFs, clear saturation of peak counts are not achieved at the recommended sequencing depth (20 million of mapped reads) and on average only about 50% of all potential peaks are identified for typical TFs at the recommended sequencing depth (Genome Res. 2012 Sep;22(9):1813-31.). Thus ChIP-seq data represents a random sampling of all potential TF binding sites, and of substantial bias because stronger peaks are often detected first. Moreover, ChIP-seq data is known to have reproducibility issues and the ENCODE guideline suggested that reproducible sets should display > 75% overlap in identified peaks, which further suggests that ChIP-seq is just a random sampling of all potential binding sites. Taken together, ChIP-seq data for the 3 TFs, even performed in the same cell

line, will not be able to reveal all genomic regions co-occupied by them but just represents a biased sampling of all potential binding sites.

In light of those observations, we could consider ChIP-seq data from different cell lines for the same TF just represent different samplings of all potential binding sites of the TF, which are of variable sampling sizes and with different degrees of bias.

Dealing with samples of limited size and with bias is a common issue in statistics and well-established techniques, such as bootstrapping, have been developed to generate robust results by resampling the data. Hence, we analyzed the ChIP-seq data of the 3TFs from Lovo and LnCAP with bootstrapping. The bootstrapped analyses did demonstrate that the 3 TFs shared a significantly higher number of microRNA targets than expected.

We have added the tissue information of the ChIP-seq data into the manuscript. We also elaborated our rationales to generate meaningful results using ChIP-seq data from different tissues with bootstrapping in the result section where the ChIP-seq analysis was first introduced.

9. The experimental validation rate of computationally predicted microRNA targets: 51% for AR, 54% for HOXC6 and 73% for NKX2-2 (Table S1, Fig. S3E) is rather low. A validation rate close to 50% is like flipping a coin.

The validation rate is determined from how many predicted targets could be experimentally validated and a validation rate close to 50% is actually far better than flip coins. It is important to note that a “flip coin” predictor is a random predictor, which will have an AUC (Area Under receiver operating characteristic Curve) of 0.5, rather than a validation rate of 0.5.

To correctly interpret the number, it is crucial to consider the prediction process. Let's use HOXC6 as an example. The prediction started from all human miRNAs, currently number around 1,800 (counting -3p and -5p as one as we did in the calculation of validation rates). Our method then predicted a subset of miRNAs (13) to be regulated by HOXC6. Seven out of the thirteen targets can be experimentally validated, which gives a validation rate of 53%. It is not known exactly how many miRNAs are true targets of HOXC6. However, the miRNA-seq data suggested that 94 miRNAs are regulated by HOXC6, and we can use this number as an approximation of the true number of miRNAs regulated by HOXC6. Now, let's assume that our predictor is a random predictor. In other words it predicts whether a target is regulated by HOXC6 by flip a coin. We then make predictions using this random predictor and take the top 13 predicted targets for experimental validation. Because the prediction is random, the predicted target are all have 50% chance to be correct regardless of their rank in the list of predicted targets, which is equal to an AUC = 0.5. Hence, we expect that $13 \cdot (94/1800) = 0.68$ target could experimentally validated, which give a validation rate of 5% ($0.68/13$). Using similar procedure, we could estimate that the random predictor will deliver a validation rate of 12% for AR and 7% for

NKX2-2. In summary, the validation rates for a TF by a random predictor is simply the number of true targets divided by the number of all miRNAs.

We concur that the validation rate may not be an appropriate metric to deliver an unbiased measure of the predictor's performance. However, as we have discussed above, the number of true miRNA targets of the master TFs is not known, so metrics such as sensitivity or specificity could not be reliably estimated. Taken together, we think validation rate is a more appropriate measure given the lack of information. We have modified the manuscript to point out that the validation rates are significantly better than random predictors and added the procedure to estimate validation rates of random predictors into the materials & methods section.

Minor comments:

1. Throughout the manuscript/figures, wherever RWPE1 cells were used, the control condition is labelled "RWPE1" instead of "control vector" or "control siRNA" etc. However, the treated conditions are labeled with the actual treatment X, frequently without "RWPE1 + X", creating confusion as to what is treatment and what is the cell line model.

We have corrected this confusion by changing all control conditions and treatments to the "RWPE-1 + X" format as suggested by the reviewer.

2. Several typos are present throughout the manuscript, e.g. FigS1 "Mormal".

We have thoroughly examined the manuscript and corrected typos suggested by reviewer 1 and 3.

3. In the Abstract: "Androgen-ablation therapies, which are the standard treatment for metastatic prostate cancer, invariably succumb to acquired resistance". The verb succumb usually refers to patients. For the therapies, in this context, I would suggest to use the term "lead to" or "are complicated by" or "are limited by the emergence of".

We have adopted "lead to" as suggested by the reviewer.

Reviewer #2:

Minor comments:

(1) For the Fisher' test, authors conducted multiple tests. Even the authors used 0.001 as a cutoff for p value, there are still lots of false-positives. A proper control is necessary.

In the manuscript we employed Fisher's exact test for two purposes. First, the test was utilized for TF-miRNA network reconstruction. For this purpose we used FDR cutoff of 0.01 to correct for the

genome-scale multiple testing. Second, we utilized Fisher's exact test to examine the overlap between gene lists such as miRNA targets of AR, HOXC6 and NKX2-2. We agree that multiple tests were performed because pair-wise overlap among the three TFs requires 3 tests. However, the number of tests is limited and certainly not on genome-scale. Considering the strict Bonferroni correction, a p-value cutoff of 0.001 is equivalent to a corrected p-value of 0.01 for 10 tests. Thus, a p-value of 0.001 should be sufficient to control false-positives in the absence of multiple-test correction.

(2) For the modules , When taking looking the targets. Are all of them enriched with cancer hallmarks (PMID: 24747696). In general, the hypothesis (PMID: 24747696) believes that there are many ways to trigger cancer progression, but they will converge to cancer hallmarks.

This is a very insightful point. The pathway enrichment results showed that all significant pathways (FDR < 0.05) are indeed converged to cancers or hallmark cancer pathways. We have added the table showing all significantly enriched pathways as supplementary table S3 and briefly discussed the results in the manuscript.

Reviewer #3:

Main comments:

1) Overall, the number (n=3) of TFs found to be mutually exclusive drivers whose overexpression induces the aberrant expression of metastasis-associated miRNAs in metastatic prostate cancer, seems indeed rather small. I sense that the criteria followed during the computational identification of these mutually exclusive driver TFs, was rather strict and perhaps various inconsistencies detected between this study and various publicly available datasets, are due to these strict criteria applied.

We totally agree that the number of identified TFs is small, and it is very likely that other TFs could also repress miRNAs in metastatic prostate cancer. However, we purposely adopted the strict procedure, which is crucial to demonstrate the key concept of this study that TFs could repress metastatic miRNA in a mutually exclusive fashion.

Statistically, the performance of a prediction algorithm can be measure by sensitivity (also called the true positive rate, which measures the proportion of positives that are correctly identified as such) and specificity (also called the true negative rate, which measures the proportion of negatives that are correctly identified as such) (quoted from wikipedia). While a perfect algorithm can be described as 100% specific and 100% sensitive, there is usually a trade-off between the sensitivity and specificity in real world applications. Take our algorithm as an example, if we want to identify more TFs that repress miRNAs in a mutually exclusive fashion as suggested by the reviewer (increase sensitivity), we will inevitably decrease specificity such that more TFs not repressing miRNA will be predicted as such.

Consequently, experimental validation could fail to show that there exist mutually exclusive TFs repressing miRNAs in metastatic prostate cancer.

Because our key goal in this study is to establish the notion that TFs could repress miRNAs in a mutually exclusive fashion, we purposely focused on achieving high specificity. Once the concept has been established, future studies can then focus on sensitivity as suggested by the reviewer, which will be crucial to identify more TFs repressing miRNA and achieve deeper understanding of the mechanisms underlying metastasis.

Finally, as we discussed in our response to comment #4, the inconsistencies are largely stemmed from the inherent variation of the microarray platform and the small sample size of different studies profiling gene expression in cancer patients. Moreover, because our list of miRNAs demonstrated significant overlap with both the Taylor and the Sboner data set, the inconsistencies are probable not due to these strict criteria applied.

2) Did the authors try to over-express or silence simultaneously all 3 TFs in RWPE-1 cell line?

We didn't perform co-silencing experiments because RWPE-1 cells don't simultaneously express the three TFs.

During revision we simultaneously overexpressed the three TFs in RWPE-1 cell as suggested by the reviewer. We measured the expression of several miRNAs and found that those miRNAs didn't display significant change of expression comparing to RWPE-1 cells overexpressing a single TF. Hence, it appears that there is no synergy among the three TFs and a single TF is sufficient to repress the targeted miRNAs. The results from simultaneously overexpressing all three TFs have been added into the manuscript as Supplementary Figure S8E-F, and its implication has been discussed in the discussion.

3) ChIP-seq results represent just a small fraction of the number of miRNAs that are located around the peaks of all 3 TFs, while the authors report a large number of them. How far from the corresponding AR, HOXC6 and NKX2-2 binding sites were the rest of the miRNAs located?

This is a very insightful point to examine the ChIP-seq data from a different perspective. Essentially, the question is asking whether those miRNAs not co-regulated by the three TFs tend to be far away from the binding sites of the three TFs comparing to the miRNAs co-regulated by the three TFs.

To address this question, we first examined the ChIP-seq data and for every miRNA in the genome we identified the AR, HOXC6 and NKX2-2 binding sites that are closest to the particular miRNA. Next, for each miRNA we calculated the median shortest distance for the three binding sites, and used the median shortest distance as an approximation to the miRNA's distance to the nearest binding sites of all three TFs. Finally, we partitioned the miRNAs into those co-regulated by the three TFs and those

not and compared their median distance to the three TFs' binding sites. The plot comparing the distribution of the median shortest distance (density plot) is shown below. As shown in the graph, there is a tendency of the miRNAs co-regulated by the three TFs to be located closer to the binding sites of the three TFs than those miRNAs not co-regulated by the three TFs. Although we observed the tendency, the p-value is only borderline significant (p-value = 0.029). Thus results from this analysis are consistent with other analyses but provide a very limited support of our conclusions. We think it is not justified to be included into the manuscript and chose to present the results in our responses to reviewer's comment.

4) The small number (n=4) of overlap in differentially expressed (DE) miRNAs among the Taylor & Hart datasets and the 3 TFs' targets, is rather alarming. A similar workflow of judging which miRNAs are differentially expressed should have been taken into careful consideration. Of course, the authors have solved this problem by querying Pubmed for publications that provide evidence that their miRNAs are associated with prostate cancer metastasis. However, this type of evidence was not detected for all of their miRNAs (but only for 56% of them), and this is worrying. Furthermore, I was wondering what would the overlap with the Glinsky and Sboner

datasets be?

We agree that the number is small. However, as we pointed out in the manuscript (page 8, second paragraph), if we only consider miRNAs that display strong downregulation (fold change >4), then the overlap between the three sets is substantial in that they share 4 out of the 6 miRNAs displaying strong downregulation. Hence, the inconsistency is largely associated with miRNAs displaying smaller downregulation in metastatic prostate cancer. Because we utilized the same golden standard microarray analysis procedure for both data sets (RMA for normalization and limma for differential expression), the results suggest that the inconsistency is largely owing to the intrinsic variation associated with the microarray technology and sampling variance associated with cancer patients in different data sets, which will lead to reduced consistency in identifying differentially expressed gene in cancer samples (Evaluating reproducibility of differential expression discoveries in microarray studies by considering correlated molecular changes, *Bioinformatics* (2009) 25 (13): 1662-1668; Reliability and reproducibility issues in DNA microarray measurements, *Trends Genet.* 2006 Feb; 22(2): 101–109). Thus, limited overlap among differentially expressed genes identified by different studies in the same cancer is not uncommon.

Importantly, although the overlap among the three sets of miRNAs is small. Our list of miRNAs regulated by the three TFs demonstrated large and statistically significant overlap with both the Taylor and the Hart data set. Thus, the small overlap among the three sets of miRNAs is the result of the variations between the Taylor and the Hart data set, which only showed overlap of 5 miRNAs. Because our list of miRNAs demonstrated significant overlap with two independent, and largely inconsistent data sets, it actually provides strong evidence that our list effectively captures the miRNAs associated with metastatic prostate cancer.

Secondly, we agree that it will be more convincing if there is a higher number of miRNAs with reported association with metastatic prostate cancer. However, during the literature search we noticed that those miRNAs lacking association with prostate cancer are generally ill characterized. For example, for the 9 miRNAs without any associated publication in Table S2, pubmed queries using only the names of those miRNA showed that 8 of the 9 miRNAs have 2 or less publication associated with them (miR-1468, 1; miR-5008, 0; miR-561, 0; miR-659, 2; miR-1180, 0; miR-3065, 0; miR-548e, 0; miR-7705, 0). This is in sharp contrast with well-studied miRNAs such as miR-205, which is associated with 118 publications. Thus, the lack of literature doesn't necessarily indicate that those miRNAs are not associated with metastatic prostate cancer, but is likely a result from the lack of studies of those miRNAs in general. The same scenario could probably apply to the 15 miRNAs that have been associated with metastasis in non-prostate cancers. In summary, scientific literature could be considered as a form of non-random sampling where "hot" miRNAs, such as miR-200 families, receive a lot of attention and their roles in inhibiting metastasis in multiple cancers are well-documented, and miRNAs not so hot receive little attention, even though they may turn out to be important regulators (the explosive growth of IDH1 publications may serve as a good example of a "boring" gene becomes

“hot”). Taken together, considering the limitation of scientific literature, the observations that 56% of the miRNAs have clear evidence for association with metastatic prostate cancer and 83% of the miRNA are associated with metastasis in at least one cancer provide strong evidence that majority of the identified miRNAs are indeed associated metastasis of prostate cancer.

Finally, we apologize for not make it clear that the Glinsky and Sboner data sets only measure mRNA expression and are not related to miRNAs.

5) Is there any association between AR, HOXC6 and NKX2-2 and Super-Enhancer regions detected in the prostate cancer genome? Perhaps the authors could search this using publicly available ChIP-seq data.

This is an excellent point. In contrast to combinatorial therapies that require drugs tailored to each master TFs, a single drug disrupting super-enhancer may be an elegant alternative, providing that super-enhancers are associated these master TFs. We examined published H3K27ac ChIP-seq data from LnCAP, PC3 and VCaP cells for super-enhancers associated with AR, HOXC6 and NNKX2-2. Overall, the cell line ChIP-seq data suggests that the three TFs are not broadly associated with super-enhancers. The data shows that AR is only associated with super-enhancers in VCaP cells. Although not associated with super-enhancers, HOXC6 and NKX2-2 are associated enhancers that were ranked within top 10% of all enhancers in two different cell lines, suggesting that they tend to be associated with stronger enhancers. It is important to note that tumor tissues could display substantial difference in epigenetics landscapes comparing to cancer cell lines (Ziller et al., Nature 500, 477–481 (22 August 2013)). Thus, it is still possible that the three TFs are broadly associated with super-enhancers in prostate cancer tumor tissues. Given the tremendous benefits gained if super-enhancers are indeed associated with the overexpression of those master TFs, future studies are justified to perform ChIP-seq with metastatic prostate cancer tissues to resolve this issue. The results have been added into the manuscript as Supplementary Table S5 and the implication of super-enhancers has been discussed in a new section in the discussion.

Minor comments:

1) Perhaps if the authors validated more miRNAs as TF targets by qPCR, the validation rate would be higher than 50%. I think that the number (n=6) of miRNAs chosen for qPCR validation in page 7 is rather low, compared to the larger number of co-regulated miRNAs (n=54).

This comment is related to the comment #9 raised by reviewer #1. As we discussed in our response to that comment, the validation rates are actually very good and comparable with the state-of-the-art network reconstruction algorithms.

For the number of qPCR validation, we would like to point out that additional qPCRs were presented in supplementary Figure S3B-D. We apologize for the confusion because it is not possible to put all qPCR results in one graph owing to space limitations. Summarizing all results, we carried out qPCR for 16 AR regulated miRNAs (48% of 33 predicted targets), 11 HOXC6 associated miRNAs (85% of 13 predicted targets) and 9 NKX2-2 associated miRNAs (60% of 15 predicted targets).

2) Perhaps it would be worthy if the cell culture-related experiments were repeated with another benign prostate cell line, apart from RWPE-1.

We agree that it will be helpful to have the cell culture-related experiments repeated with another benign prostate cell line. Unfortunately, within the allowed time frame (three months) for revision, it is not possible to carry out this task.

In China, the cell bank operated by the Chinese Academy of Sciences (CAS) is the only credible source to get cell lines because they routinely perform STR assays and guarantee the identity and quality of provided cells. Unfortunately, RWPE-1 is the only benign prostate cell line in the cell bank of CAS. We also asked several groups within Chinese Academy of Sciences and Fudan University, and had no luck in obtaining another cell line. Finally, ordering from ATCC is also not feasible within the allowed time frame. The authorized ATCC distributor in China is Beijing Zhongyuan Ltd. The estimated delivery time is around 3 months owing to the administrative procedures we have to go through and the time required to clear custom for live cell orders. Slow delivery of scientific materials is one of the challenges to face while doing scientific research in China. For example, the LSD1 antibody we used in the revision took three weeks to be delivered. We apologize for not being able to carry out this task owing to issues beyond our control.

3) Line 149-15: "RWPE-1 was preferred over prostate cancer cell lines because the dysregulation of metastasis associated microRNAs have already occurred in those cancerous cell lines". Do the authors mean that dysregulation of metastasis associated microRNAs has been reported previously in this cell line (by the corresponding cited works)?

Yes. The cited works reported that those miRNAs are repressed in those prostate cancer cells. We have modified the sentence to make it clear that because metastatic miRNAs are already repressed in those prostate cancer cell lines, it is not possible to test the repressive effects of the three TFs on metastatic miRNAs. Consequently, we used RWPE-1, which still expresses high-level of metastatic miRNAs.

4) Fig S1: Please replace the words "MORMAL" and "Hsa-miR-205" with "NORMAL" and "hsa-miR-205", respectively (vertical axis) in the legend.

The typos have been corrected.

5) Line 140: "Fig. S2B-C" should be mentioned.

Fig. S2B-C has been added.

6) Line 141: "no significant overlap was observed near coding genes". Figure S4E represents the overlap between all genes (coding and non-coding) and is misleading with the above sentence. Judging from Figure S4E, it seems that there's a large number of overlapping near coding genes (1,631 genes). The authors could represent here the overlap between just the coding genes that are associated with AR, HOXC6 and NKX2-2 binding events derived via bootstrapping of published ChIP-seq data (and exclude the non-coding genes).

As suggested by the reviewer, we have carried out bootstrapping analysis using only coding genes and updated Figure S4E. Similar to the old results using all genes, although the overlap is large among coding genes, the overlap is expected and not statistically significantly higher than the overlap expected from two independent gene lists of the same size.

7) Lines 184-186: What microarrays were used for further validation of the mRNA expression changes induced by overexpressing AR, HOXC6 and NKX2-2? It is not mentioned in the Materials & Methods.

Corresponding information has been added to Materials & Methods as a new section "MicroRNA sequencing and microarray analysis".

8) Line 176: Please cite the published ChIP-seq data used for bootstrapping analysis.

Reference has been updated to address reviewer's comments.

9) Line 191: Please cite the Taylor data set used.

The reference has been added to the manuscript.

10) Line 196: Please replace "has-miR-548c-3p" with "hsa-miR-548c-3p".

The typo has been corrected.

11) Line 417: Please replace "Has-miR-200c-3p" with "hsa-miR-200c-3p".

The typo has been corrected.

12) Line 488: "...cells were incubated at 37oC for 3 h."

The typo has been corrected.

13) Figure 4: Please revert the axes of the Scatter plot showing the mutually exclusive overexpression between HOXC6 and NKX2-2, accordingly, between subfigures A and B.

We have updated Figure 4 according reviewer's comments.

14) The role of epigenetic mechanisms affecting dysregulated AR gene expression could be mentioned in the discussion.

We have added a paragraph in discussion to discuss the role of epigenetic mechanisms and LSD1 (as point out by reviewer #1) in affecting dysregulated miRNA expression in prostate cancer.

REVIEWERS' COMMENTS:

Reviewer #1 (Remarks to the Author):

Comments

- Regarding my original comment #1 (on the use of the term “mutually exclusive relationship”), the authors have responded that it “should be interpreted from a statistical perspective in that as far as two or more genes demonstrate statistically significant tendency towards mutual exclusivity”. I had interpreted the mutual exclusivity in its biological sense (= absolutely no sample overlap), which could potentially be related to synthetic lethality (two incompatible events, as the authors discuss in their response to my original comment #5). In any case, to clarify this, I think the authors could make clearer in their text that their definition of mutual exclusivity is for “statistically significant tendency towards mutual exclusivity” and not the absolute/no-overlap definition.

Reviewer #2 (Remarks to the Author):

My concerns have been addressed.

Reviewer #3 (Remarks to the Author):

The authors have provided a point-by-point rebuttal letter to all the referee's comments and have ameliorated the manuscript, to a significant level. Their replies are satisfactory and I believe that the revised version of their work deserves to be published.

REVIEWERS' COMMENTS:

Reviewer #1 (Remarks to the Author):

Comments

- Regarding my original comment #1 (on the use of the term “mutually exclusive relationship”), the authors have responded that it “should be interpreted from a statistical perspective in that as far as two or more genes demonstrate statistically significant tendency towards mutual exclusivity”. I had interpreted the mutual exclusivity in its biological sense (= absolutely no sample overlap), which could potentially be related to synthetic lethality (two incompatible events, as the authors discuss in their response to my original comment #5). In any case, to clarify this, I think the authors could make clearer in their text that their definition of mutual exclusivity is for “statistically significant tendency towards mutual exclusivity” and not the absolute/no-overlap definition.

Response: We have modified the first paragraph of the result section to clearly state that the mutual exclusivity is for “statistically significant tendency towards mutual exclusivity” as suggested by the reviewer.

Reviewer #2 (Remarks to the Author):

My concerns have been addressed.

Reviewer #3 (Remarks to the Author):

The authors have provided a point-by-point rebuttal letter to all the referee's comments and have ameliorated the manuscript, to a significant level. Their replies are satisfactory and I believe that the revised version of their work deserves to be published.